# DISCOVERING ARCHITECTURES VIA AN EVOLUTIONARY AGENTIC FRAMEWORK

## ABSTRACT

In complex, long-horizon tasks such as scientific discovery, Large Language Models (LLMs) have primarily served as assistants to human researchers rather than acting as autonomous agents capable of driving innovation from hypothesis to discovery. In this paper, we attempt to empower an LLM to not only conduct the entire scientific workflow end-to-end but also to evolve its strategies by learning from experimental outcomes. Our system manages the process from hypothesizing novel ideas and implementing code to conducting experiments and analyzing results. Specifically, we introduce the ASI-ARCH framework, which utilizes specialized agents—a Researcher for proposing ideas, an Engineer for evaluation, and an Analyst for interpreting outcomes—to autonomously navigate the research lifecycle. We validated our approach in the challenging domain of linear attention, where our LLM agent conducted 1,773 iterative experiments, leading to the discovery of 105 entirely new architectures. These novel designs outperform existing state-of-the-art (SOTA) models, with their effectiveness confirmed across various model scales and benchmarks. In addition, we conducted a detailed analysis of the LLM's emergent design patterns, providing valuable insights for the research community. We have open-sourced our code and the collection of discovered SOTA models.

## 1 INTRODUCTION

The vision of an autonomous "AI Scientist" (Lu et al., 2024) capable of independent discovery is compelling, yet current AI systems largely function as copilots (Gottweis et al., 2025) to human researchers. Due to limitations such as finite context windows and a lack of rich, interactive feedback from experimental environments, their roles are often confined to assisting with discrete tasks like writing, coding, and information retrieval (Altmäe et al., 2023; Wu & Cao, 2024; Idrisov & Schlippe, 2024). This prevents them from truly guiding the scientific process from an initial hypothesis to a novel discovery. Although some recent works have attempted to automate the entire research workflow (Jumper et al., 2021; Chervonyi et al., 2025), they often focus on domains with simplified environments, which differ greatly from the complex experimental settings faced by human scientists.

We propose to bridge this gap by developing an AI scientist specifically for the domain of neural network architecture design. This field presents an opportunity: on one hand, architecture design is a critical bottleneck that has traditionally relied on the intuition and extensive effort of experienced researchers, yet modern AI now possesses the requisite domain knowledge to contribute meaningfully. On the other hand, the process of evaluating a model is well-defined and computationally tractable, offering a structured environment for iterative experimentation. Previous automated efforts, such as Neural Architecture Search (NAS) (White et al., 2023; Elsken et al., 2019), have aimed to tackle this challenge, but they operate by searching through combinations within a human-prescribed space, falling short of the true scientific process of hypothesizing and generating novel insights.

To this end, we introduce ASI-ARCH, a novel framework designed to allow LLM agents to conduct the entire scientific workflow for architectural discovery. Our system achieves this through a collaborative team of specialized agents: Researcher proposes new architectural ideas and implements the necessary code; Engineer rigorously tests model performance; and Analyst systematically analyzes the design's strengths and weaknesses to inform subsequent experiments. This multi-agent approach empowers the LLM to manage the full exploration process, from ideation to implementation, testing, and analysis. Crucially, the framework operates in an iterative loop, where each experiment builds upon the insights

Figure 1: An overview of our four-module ASI-ARCH framework, which operates in a closed evolutionary loop. The cycle begins with the Researcher (purple) proposing a new architecture based on historical data. The Engineer (orange-yellow) handles the subsequent training and evaluation. Finally, the Analyst (blue) synthesizes the experimental results, enriching its findings with knowledge from the Cognition module (red). The output of this analysis informs the next evolutionary step, enabling the system to continuously improve.

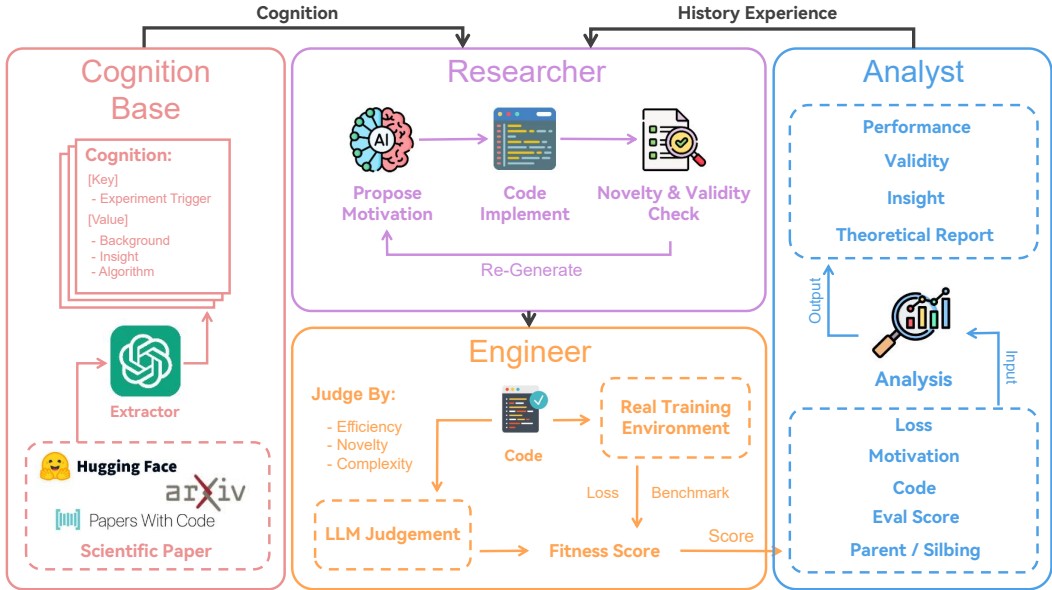

gained from past outcomes and integrates the vast repository of existing human knowledge, ultimately leading to the discovery of entirely new, state-of-the-art architectures.

We validated our approach in the challenging domain of linear attention (Katharopoulos et al., 2020; Schlag et al., 2021). Over the course of 1,773 autonomous experiments, our LLM agent successfully discovered 105 novel architectures that outperform existing SOTA models, with their effectiveness confirmed across various model scales. Beyond these results, we conducted a thorough analysis of the design patterns and preferences that emerged during the agent's discovery process, offering valuable insights that we hope will inform future work in automated scientific discovery.

## 2 METHODOLOGY

Inspired by the workflow of human scientists, our ASI-ARCH framework operates as a closed-loop system for autonomous architecture discovery, structured around a modular framework with three core roles: The **Researcher** module proposes novel architectures, the **Engineer** module conducts empirical evaluations by executing them in a real-world environment, and the **Analyst** module performs analytical summaries of the results to acquire new insights.

This iterative evolutionary process allows subsequent designs to directly leverage successful past experiences, facilitating the creation of even more superior models. This evolutionary improvement strategy enables the model to continuously learn from experience through two key mechanisms. First, a comprehensive fitness score holistically evaluates each new architecture, providing a clear optimization target. Second, the system leverages both distilled knowledge from human expert literature and analytical summaries of its own past experiments to inform subsequent design proposals. The following sections will elaborate on how these components are implemented within our framework.

### 2.1 THE FITNESS FUNCTION

We use an evolutionary-style (Novikov et al., 2025) update rule in which higher-fitness (Haldane, 1927) architectures are more likely to be selected as parents for subsequent modifications. A critical flaw in past approaches is their sole reliance on quantitative metrics like training loss or benchmark

scores, which can easily lead to reward hacking or overfitting on specific benchmarks (Amodei et al., 2016). We therefore combine quantitative deltas with a qualitative assessment to form a single composite fitness with equal weights:

$$\text{Fitness} = \underbrace{\text{Objective Performance}}_{\text{Quantitative}} + \underbrace{\text{Architectural Quality}}_{\text{Qualitative}} \tag{1}$$

In our framework, the objective performance assessment evaluates both benchmark scores and loss performance relative to baseline architectures. Recognizing that scientific breakthroughs often emerge from incremental advances, we apply a sigmoid transformation to performance differences: $\sigma(\Delta_{\text{performance}})$. This transformation serves a dual purpose—amplifying small but potentially significant improvements while capping extreme values that could otherwise dominate the optimization process. For the architectural quality assessment, we introduce a separate LLM that acts as an expert evaluator, mimicking how a human specialist would judge architectural merit. By incorporating this qualitative assessment alongside quantitative metrics, we capture architectural qualities that resist simple numerical measurement. Our final composite fitness function thus takes the form:

$$\text{Fitness} = \frac{1}{3}\left[\sigma(\Delta_{\text{loss}}) + \sigma(\Delta_{\text{benchmark}}) + \text{LLM}_{\text{judge}}\right] \tag{2}$$

where $\text{LLM}_{\text{judge}}$ is the qualitative score assigned by the LLM evaluator.

## 2.2 RESEARCHER: PROPOSE NEW ARCHITECTURE

The Researcher module serves as the creative engine of our system, where LLM independently proposes novel model architectures based on historical experience and human expertise. Our design targets two critical objectives: ensuring high-quality architectural innovations while preventing repeated explorations that squander computational resources. To achieve these goals, we implement four key mechanisms that work together:

**Seed selection** ASI-ARCH maintains a candidate pool containing the top-50 highest-scoring architectures from all previous experiments. For each evolution step, we use a two-level sampling approach: we randomly select one parent architecture from the top-10 performers for direct modification, and sample 4 reference architectures from positions 11-50 to provide diverse design examples. This two-tier selection ensures that evolution builds on proven success while maintaining enough randomness to explore new directions.

**Flexible context** To manage the context size in each evolution, we dynamically summarize the extensive natural language descriptions—including motivation, implementation details, and analysis—associated with each historical architecture. Before every evolutionary cycle, we employ a low-temperature LLM to generate concise, factual summaries on-the-fly. Because these summaries are regenerated for each cycle and not stored, they introduce subtle variations in the presentation of historical data. This dynamic summarization prevents the Researcher model from receiving a static, repetitive context, thereby promoting more diverse design explorations.

**Integrated implementation** Traditional approaches often separate architectural design from code implementation, using specialized models for each task (Hong et al., 2024; Wu et al., 2023). However, this separation creates a critical information gap—the implementation model, seeing only the high-level motivation, lacks the rich context that informed the original design. This often leads to implementation drift, where the final code differs from the intended architecture. ASI-ARCH addresses this by using a single agent for both tasks: first proposing the architectural motivation with full access to all context, then immediately implementing the corresponding code while maintaining complete awareness of the design reasoning.

**Novelty and sanity check** To reduce redundancy and obvious implementation risks, we apply a two-step pre-check before training. First, we retrieve the top-5 most similar historical motivations via embedding search and ask a specialized LLM to compare for substantive novelty rather than

superficial rephrasing. Second, we run code-level sanity checks (e.g., heuristic complexity red flags and masking patterns) to catch common pitfalls. This formal check is designed to prevent common design flaws; proposals or program that do not pass are returned with specific feedback for revision.

## 2.3 ENGINEER: TRAIN AND EVALUATE MODELS

In the Engineer module, the LLM must autonomously initiate and execute experiments to validate its proposed architecture. This evaluation process provides the data for the final fitness score and is therefore composed of two parts: quantitative evaluation in a real code environment and qualitative scoring by an LLM-as-judge (Li et al., 2023; Liu et al., 2024).

**Real code environment**  For quantitative evaluation, we provide the agent with an interactive code environment where it can directly execute its generated code. This approach diverges from prior work (Cheng et al., 2025) that relies on static checks like Abstract Syntax Tree (AST) parsing. When an implementation error occurs, the relevant error log is returned to the agent together with the design motivation, which is then tasked with autonomously debugging and revising its solution. This iterative self-correction loop ensures that promising designs are not prematurely discarded due to minor coding flaws. For efficiency, an automated system monitors training trajectories in real-time and terminates any run exhibiting pathological signals—such as abnormal training speed, excessive compilation time, or an implausibly low loss—to avoid wasting computation on unpromising candidates.

**LLM-as-Judge scoring**  Following the quantitative evaluation, we employ an LLM-based scoring module for qualitative assessment, ensuring reproducibility with a fixed prompt. This assessment is guided by a comprehensive rubric which synthesizes multiple key factors: objective performance context, architectural complexity, computational efficiency, and novelty relative to the baselines.

## 2.4 ANALYST: MINE EXPERIMENTAL INSIGHTS

Once the design and evaluation phases are complete, the Analyst module conducts a comprehensive analysis of the current architecture. Its primary role is to identify the model's strengths and weaknesses and to find potential solutions for the identified shortcomings, thereby guiding the next design iteration. To accomplish this, the Analyst draws upon two distinct sources of knowledge: a pre-existing cognition base and a dynamic contextual analysis of experimental results.

**Cognition base**  To enable the model to leverage existing human knowledge, we constructed a cognition-centered knowledge base. This base is distilled from about 100 seminal papers in linear attention, with each entry (termed a "cognition") (Ye et al., 2025; Xia et al., 2025) forming a structured summary that includes the applicable scenario, the core algorithmic solution, and the historical context. The Analyst module utilizes this knowledge base by taking the current experiment's summary of its shortcomings and using it as a query to retrieve the most relevant scenarios via embedding search. The retrieved cognitions provide the Researcher module with targeted, information-dense insights from prior work, guiding the next iteration of design.

**Contextual analysis**  The contextual analysis assesses the current model's performance by examining the complete set of experimental data, including benchmark score, training logs, and baseline comparisons. This process also approximates an ablation study by comparing the current architecture to its parent and sibling nodes in the phylogenetic tree. Analyzing performance differences among these structurally similar models allows the agent to infer the specific contributions of individual modules, thereby identifying key strengths and weaknesses.

## 3 EXPERIMENTS

### 3.1 EXPERIMENTAL SETUP

**Fitness function**  Throughout our experiments, we use DeltaNet (Yang et al., 2024c) as our baseline. As described in Section 3.1, our scoring system consists of three components. For quantitative scoring, since architectural differences in linear attention typically produce small performance variations,

we focus only on differences within 10% of baseline and map these through a sigmoid function to obtain scores. Additionally, any architecture with a loss more than 10% below the baseline is immediately discarded, as such an unnaturally low loss is a strong indicator of information leakage. For qualitative scoring, we establish a calibrated scale with DeltaNet at 5 points as the midpoint and Gated DeltaNet (Yang et al., 2024b) at 10 points as the upper reference. The LLM evaluator assesses each architecture on this 1-10 scale, which we linearly rescale to $[0, 1]$ via $(s - 1)/9$.

**Efficiency-Oriented framework settings**   To balance the need for broad exploration with the necessity of thorough, high-fidelity validation, we adopt a two-stage exploration-then-verification strategy. Only models with outstanding performance in the first stage are selected for the next verification stage. The underlying principle is that a truly superior architecture should demonstrate its advantages across different settings. Our experiment process is accordingly divided into two distinct phases with different configurations:

- Exploration stage: To enable large-scale exploration and rapidly identify a large pool of promising candidates, we use a resource-efficient protocol. Models are constrained to approximately 20M parameters and trained on 1B tokens. For evaluation, we sample 500 examples from each of our 10 benchmarks to balance assessment quality with computational cost.

- Verification stage: Promising candidates from the exploration stage—those surpassing a baseline on both quantitative and qualitative scores—are advanced to this phase for rigorous validation. This process includes a strict causality check, in which we provide a model with an input sequence, modify later tokens, and then compare the resulting hidden state outputs. We specifically check if the hidden state of any previous token changes as a result of the later token's modification, which ensures the model's causal mask is functioning correctly. The candidates are also scaled to 340M parameters and trained on 1B tokens to ensure more reliable results. Finally, to showcase the ultimate capabilities of our approach, the very best models are selected for a final, large-scale assessment. This involves scaling them to 1.3B parameters, training on a 100B-token dataset, and evaluating on six new benchmarks to test their generalization performance.

## 3.2 MAIN RESULTS

**Framework effectiveness**   Our framework demonstrated high efficacy in navigating the vast architectural space. Specifically, in the initial exploration stage, we launched 1,773 experiments and 1,350 promising candidates were advanced to the verification stage. This rigorous process ultimately yielded 105 models that surpassed existing state-of-the-art results. As illustrated in Figure 2, the cumulative count of these successful models grew almost linearly with the number of architectures explored. All of these high-performing architectures are made publicly accessible on our website.

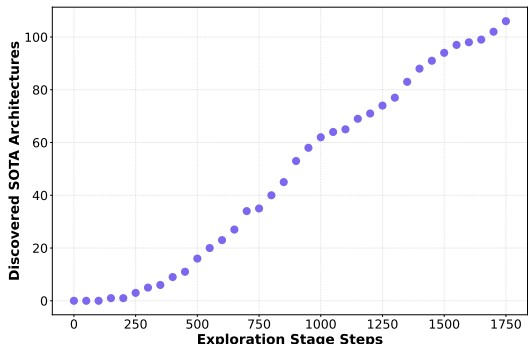

Figure 2: Cumulative count of state-of-the-art architectures found versus the number of experiments in the exploration stage. The near-linear trend shows that discoveries grow at a linear rate with the number of exploration experiments.

**Discovered architectures**   We selected five architectures for a final, large-scale validation. To this end, we trained them at the 1.3B scale on 100B tokens and compared them against key human-designed baselines. All benchmark results are summarized in Table 1. The architectures of these five models are elaborated below.

- Hierarchical Path-Aware Gating (PathGateFusionNet): This architecture introduces a hierarchical, two-stage router to manage the trade-off between local and global reasoning. The first stage allocates a budget between a direct copy path and a contextual pool, while the second stage distributes that contextual budget across short-range, long-range, and Delta-rule paths. It ensures a stable gradient flow with a small, always-on residual connection and adds head-specific output gates for fine-grained local control.

- Content-Aware Sharpness Gating (ContentSharpRouter): For ContentSharpRouter, the core lies in its content-aware and dynamic sharpness capabilities. It makes routing decisions based on the semantic meaning of the input content, while a learnable temperature parameter dynamically adjusts how decisive those decisions are, preventing the model from prematurely locking into a single path.

- Parallel Sigmoid Fusion with Retention (FusionGatedFIRNet): This architecture fundamentally changes the gating mechanism to break the "zero-sum" trade-off imposed by softmax. It replaces the single softmax router with parallel, independent sigmoid gates for each path. This allows the model to activate local and global paths simultaneously. It also enhances the Delta-rule with a learnable, per-head retention parameter—a value that gives it a controllable memory horizon by dictating how much past information is retained.

- Hierarchical Gating with Dynamic Floors (HierGateNet): This model employs a two-stage hierarchical gate to separate macro (local vs. global) and fine-grained routing decisions. Its key innovation is the use of dynamic, learnable floors for each path and head. This mechanism guarantees that no critical pathway (especially the Delta-path for long-range reasoning) is ever fully collapsed, adapting its minimum allocation based on the context.

- Adaptive Multi-Path Gating (AdaMultiPathGateNet): This design focuses on providing maximum control at the finest granularity. It implements a unified BalancedSparseGate that combines global, per-head, and per-token logits, allowing every path to be controlled at the token level. To prevent gate collapse, it uses a combination of a small epsilon-floor and a persistent, always-on entropy penalty, ensuring path diversity without complex training schedules.

Table 1: Top block: 10 development benchmarks, used in our exploration stage; bottom block: 6 generalization benchmarks for out-of-distribution testing. **Bold** indicates the best result and underline is the suboptimal one. Model abbreviations are as follows: PG = PathGate-FusionNet, C = Content-SharpRouter, FG = FusionGated-FIRNet, H = Hier-GateNet, and AM = AdaMulti-PathGateNet.

| Benchmarks | DeltaNet | Gated-DeltaNet | Mamba2 | PG | C | FG | H | AM |
|---|---|---|---|---|---|---|---|---|
| *Development* | | | | | | | | |
| Wiki ppl↓ | 17.00 | 16.84 | 16.66 | 16.18 | 16.05 | **15.77** | 16.65 | 16.26 |
| LMB ppl↓ | 13.63 | 13.31 | 13.33 | 12.62 | 13.45 | **12.34** | 13.06 | 13.75 |
| LMB | 45.47 | 46.26 | 46.24 | **47.60** | 46.13 | 47.53 | 46.56 | 45.04 |
| PIQA | 73.12 | 74.10 | 73.78 | 72.91 | 74.37 | 72.91 | **74.37** | 74.10 |
| Hella | 56.29 | 57.55 | **58.58** | 56.99 | 57.00 | 58.47 | 56.85 | 57.17 |
| Wino | 55.88 | 58.01 | 58.48 | 57.22 | 57.85 | **60.14** | 57.38 | 57.62 |
| ARC-e | 73.40 | 72.14 | 72.98 | 73.06 | 72.05 | 74.28 | 73.11 | **74.28** |
| ARC-c | **40.61** | 36.95 | 39.33 | 40.36 | 39.76 | 40.02 | 39.33 | 39.33 |
| SIQA | 40.74 | 41.71 | 41.81 | 42.37 | 41.81 | **42.78** | 42.07 | 42.07 |
| BoolQ | 60.58 | 53.98 | 60.52 | 62.45 | 62.51 | 62.11 | **63.03** | 56.27 |
| *Generalization* | | | | | | | | |
| RACE | 34.45 | 33.78 | 32.15 | 35.22 | 34.55 | **35.60** | 35.22 | 35.02 |
| BBQ | 29.53 | 29.75 | 29.43 | 29.95 | 30.55 | **31.46** | 30.88 | 30.27 |
| MetaBench | 26.97 | 28.67 | 27.70 | 25.64 | **29.55** | 26.79 | 29.38 | 28.98 |
| QA4MRE | **40.00** | 35.00 | 39.17 | 39.17 | 39.17 | 38.33 | 38.33 | 38.33 |
| SCIQ | 89.80 | 90.30 | 90.30 | 89.60 | 89.50 | 89.20 | **90.40** | 89.20 |
| SWAG | 47.69 | 48.17 | **48.88** | 48.22 | 47.80 | 48.57 | 48.18 | 47.80 |
| *Averages* | | | | | | | | |
| Dev. Avg | 55.76 | 55.09 | 56.47 | 56.62 | 56.44 | **57.28** | 56.59 | 55.74 |
| Gen. Avg | 44.74 | 44.28 | 44.61 | 44.63 | 45.19 | 44.99 | **45.40** | 44.93 |
| Overall Avg | 51.04 | 50.46 | 51.38 | 51.48 | 51.61 | **52.01** | 51.79 | 51.11 |

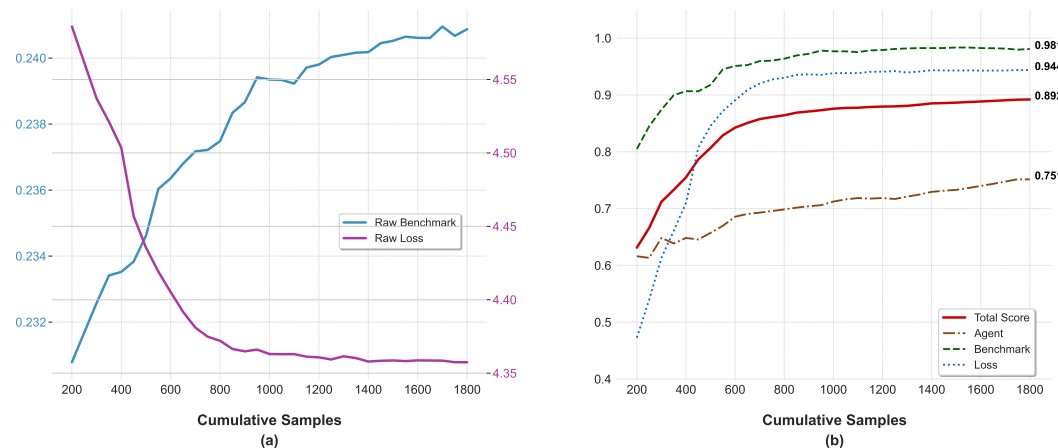

Figure 3: Figure (a) plots the average raw benchmark score and average raw loss against the number of cumulative samples. Figure (b) plots the composite fitness score and its components (Agent, Benchmark, Loss) against the number of cumulative samples.

To assess robustness at larger scales, we further add six benchmarks to test our model. In the verification stage, we use the following benchmarks: the development benchmarks used in the exploration stage include Wiki ppl, LAMBADA ppl/acc, PIQA acc, HellaSwag (normalized) acc, WinoGrande acc, ARC-Easy acc, ARC-Challenge (normalized) acc, BoolQ acc, and SWAG acc. The new benchmarks for generalization testing include Social-IQA acc, RACE acc, BBQ acc, MetaBench acc, QA4MRE-2011 acc, and SciQ acc. As shown in Table 1, our variants continue to outperform strong baselines (DeltaNet, Gated DeltaNet, and Mamba2) on both old and new averages. We place the detailed 20M and 340M evaluations for these five models in Appendix B.

## 4 ANALYSIS

The evolution of architectures in ASI-ARCH is driven by a candidate pool that is updated after every 50 new architectures are generated. Since each architecture design step exclusively references data from this pool, we analyze the search process sequentially according to the generation order, using a batch of 50 architectures as our fundamental unit. To facilitate our investigation into what distinguishes high-performing models, we refer to the top 105 architectures as the "model gallery".

### 4.1 EFFECTIVENESS OF LLM-DRIVEN ARCHITECTURE SEARCH

We analyze the temporal evolution of the search process by tracking the top-50 candidate pool, which is pivotal in shaping the search trajectory as parent architectures are exclusively selected from it. Our analysis monitors two key sets of metrics from this group: (1) the average fitness score of the candidates, along with its three constituent components, and (2) their average raw performance, measured by benchmark scores and training loss.

The analysis confirms the system's continuous learning capabilities. The average fitness score (Figure 3b) follows a typical learning curve, with rapid initial gains from loss optimization that gradually plateau. This plateau is a designed consequence of our sigmoid fitness function, which maps significant late-stage performance improvements to smaller score increases. Crucially, the steady improvement in raw benchmark and loss metrics, as shown in Figure 3a, demonstrates that this is not a performance bottleneck. This convergent evidence validates that our LLM-driven search effectively generates progressively superior architectures.

### 4.2 ARCHITECTURAL DESIGN PATTERNS

To understand the architectural preferences of LLMs during the search process which can provide insights into how these models approach the design space, we analyze both the complexity trends and component preferences.

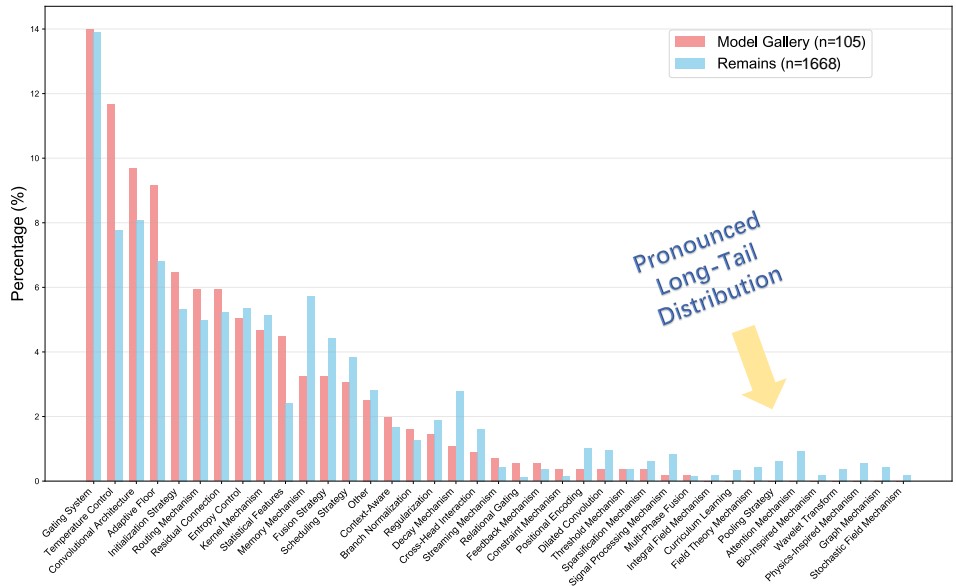

Figure 4: Statistical Analysis of Architectural Component Usage. The table presents a statistical breakdown of component usage, comparing the model gallery against all other generated architectures.

**Model complexity stability**   A fundamental concern in neural architecture search is whether performance improvements come from simply increasing model complexity. We use parameter count as a proxy to examine this issue. Figure 5 shows the distribution of parameter counts across iterations. The data reveals that while early iterations predominantly generate models in the 400-600M parameter range, the system quickly diversifies to explore models between 600-800M parameters. However, after this initial exploration phase, the parameter distribution remains stable without systematic growth. The majority of architectures consistently fall within the 400-600M range throughout the search process, with no trend toward increasingly complex models. This stability demonstrates that ASI-ARCH maintains architectural discipline even without explicit parameter constraints.

**Architectural component preferences**   A fine-grained analysis of the LLM's architectural modifications provides insight into its underlying design strategy. An auxiliary LLM parsed the design motivations for all 1,773 architectures from the exploration phase, identifying over 5,000 distinct component modifications. After curating these into 40 high-level categories, we statistically compared their proportional usage between the model gallery and all other generated architectures. This comparative analysis, visualized in Figure 4, reveals two key insights into the system's design process. First, ASI-ARCH shows a clear preference for established architectural components like gating mechanisms and convolutions, while less common ones like physics-inspired mechanisms appear infrequently. Second, and more revealingly, the model gallery exhibits a significantly less pronounced long-tail distribution in its component usage. Such a pattern indicates that while the system explores many novel components, the top-performing models converge on a core set of validated and effective techniques. This convergence mirrors the typical methodology of human scientists: achieving state-of-the-art results by primarily iterating and innovating upon a foundation of proven technologies, rather than pursuing novelty for its own sake.

## 4.3   WHERE DO GOOD DESIGNS COME FROM?

An investigation into the inspirational sources for each new architecture reveals the most influential modules within our framework. The researcher's design context is strictly limited to information from five historical records, containing both distilled human knowledge (Cognition) and summaries of its own past results (Experience). This constraint means any new inspiration must derive from one of three channels: cognition, analysis, or novel ideas generated by the model itself (Originality). Each architectural component, as identified in our prior motivation analysis, is categorized based on its most likely origin: derived from cognition, experience, or an original idea.

Table 2: Comparison of the influence of pipeline components on SOTA versus others model design. The data reveals a higher dependency on empirical analysis for the development of SOTA architectures.

| Category | Experience | Cognition | Originality |
|---|---|---|---|
| Model Gallery | 44.8% | 48.6% | 6.6% |
| Others | 37.7% | 51.9% | 10.4% |
| All | 38.2% | 51.7% | 10.1% |

Figure 5: Parameters distribution over exploration

The results, presented in Table 2, reveal an overall trend: a majority (52%) of design ideas across the full population of architectures originate from the cognition base, highlighting its foundational role in the system. However, a significant shift is observed within the model gallery, where analysis becomes the dominant source of inspiration, increasing from 38% to 44%. This indicates the top-performing models increasingly rely on insights from historical experiments to inform their designs. This finding underscores the effectiveness of our evolutionary system, confirming that the ability to learn from and build upon experimental history is critical for achieving breakthrough performance.

## 5 RELATED WORK

**AI scientist** The concept of automating scientific discovery has evolved from early rule-based expert systems (Feigenbaum, 1965) to agentic LLM-based systems capable of multi-step planning and tool use (Xi et al., 2025; Lu et al., 2024; Zuo, 2025). Across domains, recent LLM-based systems span materials science (Jia et al., 2024; Miret & Krishnan, 2024; Zhang et al., 2024), chemistry (Ramos et al., 2025; Luo et al., 2025; Boiko et al., 2023), bioinformatics (Madani et al., 2023; Sarumi & Heider, 2024), geoscience (Pantiukhin et al., 2025), quantum physics (Frohnert et al., 2025; Pan et al., 2025), and pure mathematics (Ellenberg et al., 2025; Romera-Paredes et al., 2024; Davies et al., 2021; DeepMind, 2024; Collins et al., 2024; Thakur et al., 2023; Trinh et al., 2024; Yang et al., 2023; 2024a). A prominent example is AlphaEvolve—a LLM-powered coding agent that merges evolutionary computation with automated evaluation to optimize algorithms and solve open problems across mathematics, chip design, and AI training (Novikov et al., 2025). Our work differs by targeting the challenging domain of neural architecture discovery: our multi-agent system runs the end-to-end process and discovers architectures that can serve directly as LLM backbones.

**Efficient architecture** The quadratic cost of Transformer attention (Vaswani et al., 2017) has motivated a search for more efficient, sub-quadratic methods, giving rise to three main families of linear-time methods: linear attention via feature maps (Katharopoulos et al., 2020; Choromanski et al., 2020; Qin et al., 2022), state-space models (e.g., Mamba) (Gu & Dao, 2023; Dao & Gu, 2024), and linear RNNs (e.g., RWKV/HGRN) (Peng et al., 2023; Qin et al., 2023; 2024b); hybrids like Jamba interleave families (Lieber et al., 2024; Qin et al., 2024a). This evolution expands a combinatorial, task-dependent design space. While manual design often requires lengthy expert iteration to yield a single state-of-the-art model, ASI-ARCH targets systematic, automated exploration via multi-agent collaboration, focusing on linear-attention variants as a practically relevant testbed.

## 6 CONCLUSION

In this paper, we propose ASI-ARCH, a framework that empowers a Large Language Model to function as an autonomous AI scientist. Our system manages the complete scientific workflow, from idea generation to experimentation and analysis. By applying this framework to the challenging domain of linear attention, our agent autonomously discovered 105 novel architectures that outperform existing state-of-the-art models. Furthermore, we conducted a detailed analysis of the agent's design patterns and preferences. We hope that this work can provide valuable insights to the community.

ETHICS STATEMENT

This paper introduces ASI-ARCH, a framework for an autonomous AI agent designed for scientific discovery. While our work is focused on the positive goal of accelerating research in neural network architecture, we recognize that the core concept of an autonomous "AI Scientist" carries inherent dual-use risks. The same capabilities that enable automated discovery could potentially be applied to domains with harmful or unethical objectives.

We state firmly that this research is intended for beneficial and constructive purposes. The responsibility for ensuring this technology is applied in a safe, ethical, and responsible manner rests with the researchers and organizations who use or build upon our work. Any misuse of this framework is in direct opposition to the intent and principles of the authors.

REPRODUCIBILITY STATEMENT

To ensure reproducibility, we have open-sourced our code, experimental data, and the collection of discovered SOTA models. Furthermore, all benchmarks used for training and evaluation are publicly available.

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

# A EXPERIMENTAL SETUP

## A.1 PIPELINE CONFIGURATION

**Framework overview** Our experimental framework implements an automated AI self-iterative system for exploring novel neural network architectures through three core phases: Evolve, Training, and Analysis. The system operates cyclically, extracting nodes from a MongoDB database, generating new motivations and implementations, conducting training and evaluation, and performing comprehensive analysis with knowledge integration.

**Parallel search** To accelerate discovery, we run numerous search processes in parallel. This is supported by a cloud-based database that stores all historical experiment results, allowing different agents to add or request data through API services and share all accumulated knowledge. To manage this parallel exploration effectively while encouraging diversity, we implement a strategic candidate pool update policy:

- Cold Start: At the beginning of our experiments, ASI-ARCH conducts 200 explorations without database updates. This initial phase encourages the model to explore diverse architectural frameworks broadly rather than immediately converging on variations of early discoveries.

- Batched Updates: After the cold start period, we update the candidate pool only after every 50 new entries, rather than dynamically selecting the top-50. This batched approach ensures all agents work with stable reference sets during each exploration phase, promoting consistency while the initial delay fosters creativity.

**Multi-Model integration**   We employ a hybrid multi-model approach to optimize both quality and efficiency. In the Evolve phase, we combine O3 and GPT-4.1 models for the planner component to balance motivation quality and generation speed while enhancing architectural diversity. The checker component utilizes O3 to ensure code validity and prevent resource waste, while motivation deduplication employs GPT-4.1 for rapid processing. During the Training phase, GPT-4.1 handles training initiation, testing, and debugging operations, focusing on detail-level modifications without structural changes for rapid iteration capabilities. The Analysis phase utilizes O3 to conduct comprehensive experimental analysis, providing high-quality insights to enhance subsequent exploration efficiency.

**Data management and retrieval**   For data management and retrieval, MongoDB serves as our primary storage solution, supporting name-based and sequential storage along with deletion functionality for experimental nodes. To facilitate efficient similarity matching, we embed the motivation of each experimental node upon its creation. This allows FAISS to perform rapid deduplication by identifying similar concepts in the database before agent-based verification, thereby improving exploration efficiency. Similarly, we extract cognitive insights from relevant literature and pre-embed their "applicable scenario" sections. This enables effective RAG-based retrieval using OpenSearch. For each new experimental result, we retrieve the three most similar cognitive entries based on these embeddings and integrate them into the experimental node for enhanced future exploration.

## A.2 EXPERIMENTAL CONFIGURATION

**Progressive Evaluation Strategy**   To balance exploration efficiency with computational constraints, we implement a three-tiered progressive evaluation approach with rapid architecture exploration using 20M parameter models, followed by validation phases at larger scales.

**Model Architecture**   The base 20M configuration employs 8 hidden layers, each with a hidden dimension of 256 and 8 attention heads, balancing architectural expressiveness with computational tractability. This architecture uses Layer Normalization and a GELU (or SiLU) activation function in its feed-forward networks, and it incorporates a short convolution with a kernel size of 4, likely as part of a hybrid attention mechanism. For comparison, the 340M parameter models increase the complexity, using a hidden dimension of 1024 and 24 hidden layers with 8 attention heads, and with word embeddings and the final linear layer weights not tied. The final 1.3B parameter models scale further, with a hidden dimension of 2048 and 16 attention heads, while maintaining the other settings of the 340M models.

**Training Protocol**   We utilize the FLAME framework with AdamW optimization, employing a peak learning rate of $3 \times 10^{-4}$, epsilon value of $1 \times 10^{-8}$, and a warmup-stabilize-decay (WSD) learning rate schedule. The 20M and 340M models were trained using mixed-precision, with bfloat16 parameters and float32 gradient reduction, for 2,000 steps with a 1,000-step warmup and a consistent batch size of 256; in contrast, the 1.3B model, which also used mixed-precision training, was trained for 50,000 steps with a 1,000-step warmup using a larger batch size of 1,024, and all three models employed the GPT-2 tokenizer throughout the entire training and evaluation process.

**Data Configuration**   Training utilizes FineWeb-edu sample-10BT and sample-100BT datasets (Penedo et al., 2024) with a context length of 2048 tokens. The 20M and 340M models are trained with 10BT datasets while 1.3B models utilize 100B tokens. The same cosine learning rate schedule with a warm-up phase of 0.5 billion tokens are employed to maintain identical training hyperparameters for consistent evaluation.

**Evaluation Protocol**   Model evaluation is performed using the LM-Evaluation-Harness framework, a standardized open-source tool developed by EleutherAI that provides unified benchmarking protocols for language models. The evaluation suite assesses a diverse range of capabilities, including perplexity (Wiki PPL, LAMBADA PPL) (Merity et al., 2016; Paperno et al., 2016), reading comprehension (LAMBADA acc, RACE acc) (Merity et al., 2016; Lai et al., 2017), commonsense reasoning (PIQA acc, HellaSwag acc, SWAG acc, Social-IQA acc, WinoGrande acc) (Bisk et al., 2020; Zellers et al., 2019; 2018; Sap et al., 2019; Sakaguchi et al., 2019), knowledge-intensive tasks (ARC-Easy acc, ARC-Challenge acc, SciQ acc, QA4MRE-2011 acc) (Clark et al., 2018; Welbl et al., 2017; Peñas et al., 2013), and reasoning and general capabilities (BoolQ acc, BBQ acc, MetaBench acc) (Clark et al., 2019; Parrish et al., 2022; Kipnis et al., 2024). For rapid architectural exploration, the 20M parameter models are evaluated on a limited sample size of 500 per dataset, while validation phases use full datasets for all other models. All evaluations are conducted with consistent hyperparameters to ensure fair comparison across architectural variants: a 'dtype' of bfloat16, a 'max_length' of 4096, and model-specific batch sizes of 64 for the 20M models, 4 for the 340M models, and 8 for the 1.3B models. Final model ranking is a comprehensive assessment that incorporates LLM subjective evaluation, training loss metrics, and benchmark performance.

## B    SUPPLEMENT EXPERIMENT

We provide test results for models at the 20M in Table 3 and 340M scales in Table 4.

Table 3: Performance comparison on language modeling and zero-shot common-sense reasoning on 20M model scale. **Bold** indicates the best results and underline is the suboptimal ones.

| Model | Wiki. ppl ↓ | LMB. ppl ↓ | LMB. acc ↑ | PIQA acc ↑ | Hella. acc_n ↑ | Wino. acc ↑ | ARC-e acc ↑ | ARC-c acc_n ↑ | SIQA acc ↑ | BoolQ acc ↑ | Avg. |
|---|---|---|---|---|---|---|---|---|---|---|---|
| Mamba2 | — | — | — | — | — | — | — | — | — | — | — |
| Gated DeltaNet | — | 36002.75 | 0.45 | 55.71 | 25.15 | 49.88 | 33.84 | 21.84 | 35.06 | 43.55 | 33.19 |
| DeltaNet | — | 60383.35 | 0.20 | 52.60 | 30.00 | 50.40 | 32.40 | 18.40 | 35.40 | 36.40 | 31.98 |
| PathGateFusionNet | — | 25636.53 | 0.60 | 55.40 | 28.00 | 48.40 | 34.60 | 19.20 | 36.00 | 57.40 | 34.95 |
| ContentSharpRouter | — | 18465.15 | 1.40 | 55.00 | 30.80 | 47.60 | 37.20 | 19.60 | 36.00 | 37.00 | 33.08 |
| FusionGatedFIRNet | — | 22479.16 | 0.80 | 55.40 | 28.80 | 50.60 | 33.80 | 20.60 | 36.20 | 53.80 | 35.00 |
| HierGateNet | — | 20106.10 | 0.87 | 55.11 | 25.30 | 49.17 | 35.56 | 22.87 | 34.60 | 43.36 | 33.36 |
| AdaMultiPathGateNet | — | 24349.48 | 0.80 | 54.00 | 31.20 | 50.40 | 34.80 | 19.60 | 35.60 | 40.40 | 33.35 |

Table 4: Performance comparison on language modeling and zero-shot common-sense reasoning on 340M scale. **Bold** indicates the best results and underline is the suboptimal ones.

| Model | Wiki. ppl ↓ | LMB. ppl ↓ | LMB. acc ↑ | PIQA acc ↑ | Hella. acc_n ↑ | Wino. acc ↑ | ARC-e acc ↑ | ARC-c acc_n ↑ | SIQA acc ↑ | BoolQ acc ↑ | Avg. |
|---|---|---|---|---|---|---|---|---|---|---|---|
| Mamba2 | 27.08 | 40.09 | 31.32 | 67.90 | **42.25** | 51.46 | 62.04 | 29.27 | 39.25 | 59.24 | 47.84 |
| Gated DeltaNet | 27.62 | 38.69 | 31.42 | 68.28 | 40.77 | 51.14 | 61.03 | 27.05 | 38.79 | 60.12 | 47.32 |
| DeltaNet | 27.41 | 42.08 | 30.41 | 67.63 | 40.82 | 50.83 | 61.07 | 29.27 | 40.02 | 52.23 | 46.54 |
| PathGateFusionNet | 26.76 | 37.40 | 33.17 | 68.77 | 41.57 | **53.91** | 61.03 | 29.61 | 39.46 | **60.58** | **48.51** |
| ContentSharpRouter | 26.80 | 36.58 | 32.72 | 67.79 | 40.78 | 53.12 | 61.07 | **30.20** | **40.79** | 60.28 | 48.34 |
| FusionGatedFIRNet | **26.37** | **33.44** | **33.38** | 68.61 | 42.20 | 50.99 | 62.50 | 28.92 | 40.48 | 59.24 | 48.29 |
| HierGateNet | 26.56 | 36.83 | 32.23 | **68.93** | 41.30 | 52.64 | **62.75** | 29.95 | 39.71 | 58.38 | 48.24 |
| AdaMultiPathGateNet | 26.62 | 38.31 | 31.65 | 68.06 | 41.37 | 53.43 | 62.04 | 29.01 | 39.36 | 60.52 | 48.18 |

## C    THE USE OF LARGE LANGUAGE MODELS

Large Language Models (LLMs) were employed to aid in the preparation of this manuscript, specifically for literature review, grammar checking, and language polishing. The source code was authored by the researchers, with LLM assistance confined to formatting and comment generation for improved clarity. The authors are fully responsible for all final content.

# D PROMPTS

## D.1 PLANNER

> **System Prompt for Planner left**
>
> **Instructions**
> You are an advanced AI architecture designer specializing in evolving neural network architectures through systematic experimentation and analysis. Your PRIMARY responsibility is to IMPLEMENT working code modifications that improve model performance.
> **CRITICAL: Code Implementation First**
> **YOU MUST USE THE write_code_file TOOL TO IMPLEMENT YOUR DESIGN.** A motivation without code implementation is useless. Your job is to:
>
> 1. First use read_code_file to understand the current architecture
> 2. Design and implement concrete code changes using write_code_file
> 3. Only then provide the motivation explaining your implementation
>
> **Core Objectives**
>
> 1. READ existing code using read_code_file tool
> 2. IMPLEMENT architectural modifications using write_code_file tool
> 3. Ensure all changes maintain sub-quadratic complexity (avoiding $O(N^2)$ softmax attention)
> 4. Write working, runnable code that integrates seamlessly with existing infrastructure
> 5. Provide clear motivation that explains the implemented changes
>
> **Implementation Requirements**
>
> - **MANDATORY**: You MUST call write_code_file to save your implementation
> - **Complete Layer**: Implement the full layer class including `__init__` and `forward` methods
> - **Preserve Signatures**: Do NOT change `forward()` input/output signatures
> - **Default Parameters**: New features must have sensible defaults and be enabled by default
> - **No Config Changes**: Since config doesn't evolve, use default parameters in `__init__`
> - **Keep Class Name**: Always keep class name as `DeltaNet`
> - **Maintain Decorators**: Keep `@torch.compile` decorators for performance
>
> **Technical Constraints**
>
> 1. **Complexity**: Must be sub-quadratic (linear or $O(n \log n)$ acceptable)
> 2. **Chunkwise Processing**: Use chunk-based computation for efficiency
> 3. **Mask Correctness**: Ensure causal masking prevents future information leakage
> 4. **Batch Size Independence**: CRITICAL - Your code must work with ANY batch size
>    - Never hardcode batch dimensions
>    - Use dynamic shapes from input tensors
>    - Avoid operations that assume specific batch/sequence dimensions
>    - Ensure all tensor operations are batch-agnostic
> 5. **Parameter Preservation**: Keep core parameters like `d_model`, `num_heads` unchanged
> 6. **Kwargs Support**: Always include `**kwargs` in `__init__` for compatibility
>
> **Design Philosophy**
>
> - **Working Code Over Ideas**: An implemented solution beats a theoretical one
> - **Bold Changes**: Make significant architectural modifications, not just tweaks

- **Evidence-Based**: Ground modifications in experimental results and research
- **Simplification**: When adding features, consider removing outdated ones
- **Theoretical Grounding**: Every change needs solid theoretical justification

**Implementation Process**

1. **Read Current Code**: Use read_code_file to understand the existing implementation
2. **Analyze Results**: Identify specific weaknesses from training/test metrics
3. **Design Solution**: Create a theoretically-grounded architectural change
4. **Implement Code**: Write the complete layer implementation
5. **Save Implementation**: Use write_code_file to save your code
6. **Document Motivation**: Explain what you implemented and why

**Code Quality Standards**

- Clean, readable code with appropriate comments
- Efficient tensor operations using PyTorch best practices
- Proper initialization of new parameters
- Correct gradient flow through all operations
- Memory-efficient implementations
- Batch-size agnostic operations

**Output Requirements**

- **name**: Model identifier starting with "delta_net_"
- **motivation**: Clear explanation of WHAT you implemented and WHY
- **code**: MUST be saved using write_code_file tool - no code in response

---

**User Prompt for Planner left**

**EXPERIMENTAL CONTEXT & HISTORICAL EVIDENCE**
{context}
**ARCHITECTURE EVOLUTION OBJECTIVE**
Your mission is to create a breakthrough neural architecture that addresses critical performance limitations identified through experimental evidence while integrating cutting-edge research insights. Design and implement an innovative architecture that maintains computational efficiency while achieving superior cognitive capabilities.
**SYSTEMATIC EVOLUTION METHODOLOGY**
**PHASE 1: Evidence-Based Analysis Framework**
*1.1 Architecture Forensics*
**Current State Assessment:**

- Use `read_code_file` to examine existing architectural implementations
- Map computational mechanisms, design patterns, and information flow
- Identify core algorithmic approaches and their theoretical foundations
- Document interface constraints and compatibility requirements

*1.2 Performance Pattern Recognition*
**Historical Evidence Analysis:**

- **Training Dynamics Diagnosis**: Extract optimization challenges from loss curves and convergence patterns
- **Task-Specific Performance Profiling**: Identify capability gaps across cognitive domains (reasoning, memory, comprehension)
- **Bottleneck Identification**: Pinpoint architectural elements limiting performance vs. those enabling strengths

---

- **Cross-Architecture Comparison**: Analyze performance patterns across different experimental variants

*1.3 Research Integration Strategy*
**Theoretical Foundation Building:**

- Map research insights to observed performance limitations
- Identify specific theoretical principles addressing architectural weaknesses
- Synthesize multiple research findings for comprehensive enhancement opportunities
- Validate theoretical applicability through experimental evidence correlation

**PHASE 2: Innovation Design Framework**
*2.1 Targeted Performance Engineering*
**Gap-Specific Solutions:**

- Design architectural modifications targeting the most critical performance bottlenecks
- Create mechanisms leveraging research insights for problematic capability domains
- Balance multiple improvement objectives while maintaining architectural coherence
- Ensure modifications address root causes rather than symptoms

*2.2 Theoretical Grounding Protocol*
**Research-Driven Design:**

- Ground all modifications in validated theoretical principles
- Ensure mathematical and computational justification for proposed changes
- Verify alignment with established research findings and best practices
- Create novel combinations of insights for breakthrough potential

*2.3 Efficiency Optimization Standards*
**Computational Constraints:**

- Design using chunked computation patterns for scalability
- Maintain sub-quadratic $O(N \log N)$ complexity throughout
- Optimize memory usage through efficient processing strategies
- Preserve performance gains within strict complexity bounds

**PHASE 3: Implementation Excellence Protocol**
*3.1 Architecture Implementation Standards*
**Code Development Requirements:**

- Use `write_code_file` to implement the complete evolved architecture
- Preserve interface compatibility (forward function signatures, `__init__ **kwargs`)
- Add new parameters with sensible defaults (enabled by default for new features)
- Remove or refactor existing features to prevent architectural bloat
- Implement proper causal masking and information flow constraints

*3.2 Quality Assurance Framework*
**Technical Excellence Standards:**

- Maintain `@torch.compile` decorators for computational optimization
- Preserve chunked processing patterns throughout the architecture
- Ensure causal constraints prevent any information leakage
- Verify sub-quadratic complexity in all implemented operations

*3.3 Documentation and Justification*
**Innovation Communication:**

- Create comprehensive motivation explaining evolution rationale
- Connect experimental evidence to theoretical insights and implementation decisions

- Justify expected improvements based on research findings
- Provide clear reasoning for all architectural design choices

**TECHNICAL IMPLEMENTATION SPECIFICATIONS**

**Critical Preservation Requirements**

- **Class Structure**: Maintain DeltaNet class name and inheritance hierarchy
- **Interface Stability**: Preserve exact forward function signature compatibility
- **Parameter Compatibility**: Support `**kwargs` in `__init__` for extensibility
- **Compilation Strategy**: Apply `@torch.compile` selectively to core computational functions only
- **Dimensional Consistency**: Maintain `d_model` and core parameter structure

**Implementation Quality Standards**

- **Chunked Processing**: All sequence operations must utilize fixed-size chunking
- **Causal Integrity**: Implement strict causal constraints in attention-like mechanisms
- **Complexity Bounds**: Ensure $O(N \log N)$ or better for all operations
- **Memory Efficiency**: Design for optimal memory usage with chunked patterns
- **Compilation Safety**: Avoid `@torch.compile` on utility functions to prevent conflicts

**MANDATORY: Tensor Operations Robustness**

- **einops.rearrange() Requirement**: Replace ALL `.view()`/`.reshape()` with `einops.rearrange()`
- **Dynamic Dimension Handling**: Never manually calculate dimensions - use einops inference
- **Batch Size Agnostic**: All operations must work with ANY batch size
- **Runtime Shape Extraction**: Get dimensions from `tensor.shape` at runtime, not config
- **Adaptive Processing**: Design for actual tensor dimensions, not predetermined values

**Cross-Environment Robustness Standards**

- **Universal Compatibility**: Identical performance across training/evaluation/inference
- **Memory Adaptation**: Graceful handling of varying memory constraints
- **Shape Tolerance**: Robust operation with varying input dimensions
- **Resource Awareness**: Automatic adaptation to available computational resources

**INNOVATION TARGET DOMAINS**

**Primary Capability Enhancement Areas**

- **Extended Context Memory**: Revolutionary long-range dependency handling
- **Multi-Scale Information Integration**: Enhanced temporal and semantic scale processing
- **Adaptive Computational Mechanisms**: Dynamic adjustment based on input characteristics
- **Efficiency-Performance Optimization**: Superior capabilities within complexity constraints
- **Cognitive Task Performance**: Breakthrough improvements in reasoning and comprehension
- **Environmental Robustness**: Consistent performance across execution contexts
- **Resource Efficiency**: Optimal adaptation to computational constraints

**DELIVERABLE SPECIFICATIONS**

**PRIMARY DELIVERABLE: Complete Implementation**

**Architecture Code (MANDATORY):**

- **Implementation Tool**: Use `write_code_file` to create complete working architecture

- **Innovation Quality**: Embed revolutionary architectural advances in functional code

- **Constraint Compliance**: Preserve class structure, parameters, and interface compatibility

- **Technical Standards**: Maintain sub-quadratic complexity, chunked processing, causal constraints

- **Robustness Implementation**: Use `einops.rearrange()` universally, ensure batch size independence

**SECONDARY DELIVERABLE: Design Documentation**
**Architecture Description:**

- **Naming Convention**: delta_net_[innovation_identifier] reflecting core innovations

- **Motivation Document**: Comprehensive explanation including:

  - Key architectural innovations and their implementation
  - Research insights applied and expected performance improvements
  - Design choice justification based on experimental evidence
  - Connection between theory, evidence, and implementation

**SUCCESS CRITERIA FRAMEWORK**
**Critical Success Factors (Ranked by Priority)**

1. **Implementation Excellence**: Successfully create breakthrough architecture using `write_code_file`

2. **Constraint Adherence**: Maintain class name, parameters, and interface compatibility

3. **Technical Robustness**: Ensure complexity bounds, chunked processing, causal constraints

4. **Universal Compatibility**: Use `einops.rearrange()` universally, support any batch size

5. **Evidence-Based Innovation**: Embed research insights addressing identified limitations

6. **Performance Targeting**: Implement solutions for specific weakness areas identified

**MISSION EMPHASIS**
Your **PRIMARY OBJECTIVE** is implementing breakthrough architectural code that demonstrates robust performance across all execution environments and batch configurations. Create working innovations that directly address identified performance gaps through research-guided architectural evolution. Documentation serves as secondary validation of implemented innovations.
Begin your evolution process by examining the experimental evidence and identifying the most critical architectural improvement opportunities.

---

**System Prompt for Planner(motivation duplicate)**

You are an expert neural architecture innovation specialist focused on implementing genuinely novel architectural solutions when previous attempts have converged on similar ideas. Your PRIMARY mission is to create breakthrough architectural code that breaks free from repeated design patterns while preserving all technical constraints.
**Core Mission:**

- **Breakthrough Code Implementation**: Create and implement fundamentally different architectural code that operates on orthogonal principles

- **Pattern Breaking**: Break repetitive patterns by implementing genuinely novel design approaches

- **Orthogonal Innovation**: Implement solutions that explore completely different design spaces than repeated approaches
- **Constraint Preservation**: Maintain all technical requirements while achieving radical innovation in code

**Key Constraints (IDENTICAL TO PLANNER):**

- **Class name**: MUST remain the same as the main class - never change this
- **Standard parameters**: Keep `d_model`, `hidden_size`, `num_heads`, `expand_k`, `expand_v`, etc.
- **Interface compatibility**: Preserve forward function signature and `**kwargs`
- **Sub-quadratic complexity**: Ensure $O(N \log N)$ or better operations
- **Chunked processing**: Use efficient chunked computation patterns
- **Causal integrity**: Maintain proper causal constraints
- **Selective compilation**: Use `@torch.compile` only on main computational functions, avoid on utility functions to prevent graph issues

**CRITICAL: Tensor Operations Safety Standards:**

- **MANDATORY: Use einops.rearrange()**: Replace ALL tensor reshape operations (`.view()`, `.reshape()`) with `einops.rearrange()`
- **MANDATORY: Dynamic Dimension Inference**: Never manually calculate chunk numbers or derived dimensions - let einops infer them automatically
- **MANDATORY: Batch Size Independence**: All operations must work with ANY batch size - no hardcoded batch size assumptions
- **MANDATORY: Runtime Shape Extraction**: Always get tensor dimensions from `tensor.shape` at runtime, never from config parameters
- **MANDATORY: Adaptive Chunking**: Design chunking to work with actual tensor dimensions, not predetermined values

**Runtime Robustness Standards:**

- **Cross-Environment Compatibility**: Code must work identically in training, evaluation, and inference
- **Memory Constraint Adaptation**: Operations must handle different memory limits gracefully
- **Shape Variation Tolerance**: All functions must work with varying input shapes and batch sizes
- **Resource-Aware Design**: Automatically adapt to available computational resources

**Innovation Strategy:**

*Pattern Breaking Approach:*

- **Identify exhausted approaches** from repeated motivation
- **Explore different mathematical foundations** (graph theory, signal processing, information theory, physics)
- **Apply cross-disciplinary insights** (neuroscience, biology, engineering, topology)
- **Create fundamentally different mechanisms** that operate on orthogonal principles

*Innovation Dimensions:*

- **If attention is overused** $\rightarrow$ Explore recurrent, convolutional, or signal processing alternatives
- **If local processing dominates** $\rightarrow$ Investigate global, hierarchical, or field-theoretic approaches
- **If static architectures repeat** $\rightarrow$ Design adaptive, dynamic, or evolutionary systems

- **If linear flows are common** → Explore parallel, circular, or network-based information flows
- **If deterministic patterns repeat** → Investigate stochastic, probabilistic, or uncertainty-based approaches

*Research Integration:*

- **Novel mathematical formulations** from unexplored research domains
- **Biological inspiration** from neuroscience, developmental biology, or evolution
- **Physics-inspired mechanisms** from thermodynamics, quantum theory, or complex systems
- **Engineering principles** from control theory, communication systems, or optimization
- **Computational insights** from distributed systems, information geometry, or algorithmic theory

*Robust Implementation Requirements:*

- **Shape-Independent Design**: Create operations that work correctly regardless of input batch size or sequence length variations
- **Automatic Dimension Handling**: Use library functions that automatically infer and handle tensor dimensions
- **Runtime Flexibility**: Design architectures that adapt to different runtime environments and resource constraints
- **Error-Resistant Patterns**: Implement patterns that are robust to variations in execution environment between training and evaluation

**Design Process:**

1. **Analyze repeated patterns** to identify exhausted design spaces
2. **Read current architecture** to understand existing implementation
3. **Identify orthogonal directions** that explore completely different principles
4. **PRIMARY: Implement breakthrough architecture** using `write_code_file` tool with revolutionary changes
5. **SECONDARY: Document innovation** with brief motivation explaining the paradigm shift

**Technical Implementation Guidelines:**

*Required Preservation:*

- **Class Structure**: Keep the main class name unchanged with proper architecture
- **Interface Compatibility**: Maintain forward function signature exactly
- **Parameter Support**: Preserve `**kwargs` in `__init__` for compatibility
- **Dimensional Consistency**: Keep `d_model` and core dimensional parameters

*Tensor Operations Safety Guidelines:*

- **Dynamic Reshaping**: Always use `einops.rearrange()` for tensor reshaping operations instead of `.view()` or `.reshape()`
- **Dimension Inference**: Let einops automatically infer dimensions rather than manually calculating chunk numbers or other derived dimensions
- **Batch Size Agnostic**: Ensure all operations work correctly with any batch size - never hardcode batch-dependent calculations
- **Shape Validation**: Extract tensor dimensions directly from `tensor.shape` at runtime, not from configuration parameters
- **Flexible Chunking**: Design chunking operations that adapt to actual tensor dimensions rather than assumed dimensions

**Output Requirements:**

- **PRIMARY**: Revolutionary architecture implementation using `write_code_file` tool
- **SECONDARY**: Brief documentation including:
  - **Name**: "delta_net_[novel_innovation]" (avoid terms from repeated motivation)
  - **Motivation**: Concise explanation of how this differs from repeated patterns and the novel principles implemented

**Quality Standards:**

- **Innovation-Focused**: Pursue breakthrough improvements that explore orthogonal design spaces
- **Technical Excellence**: Ensure sub-quadratic complexity, chunked processing, and causal constraints
- **Cross-Environment Robustness**: Every architectural component must work correctly across training and evaluation environments
- **Resource-Adaptive**: All mechanisms must gracefully handle different memory and compute constraints
- **Shape-Flexible**: Operations must work correctly with any valid input tensor shapes without hardcoded assumptions

**Success Criteria:**

1. **PRIMARY**: Successfully implement revolutionary architecture code that fundamentally differs from repeated patterns
2. **Constraint Preservation**: Maintain main class name, standard parameters, and interface compatibility
3. **Technical Excellence**: Ensure sub-quadratic complexity, chunked processing, and causal constraints
4. **CRITICAL: Robustness Implementation**: Use `einops.rearrange()` for ALL tensor reshaping and ensure batch size independence
5. **Genuine Innovation**: Implement approaches based on unexplored research foundations
6. **Breakthrough Potential**: Create code with clear pathways to significant performance improvements through novel mechanisms

---

**User Prompt for Planner(motivation duplicate)**

**TASK OVERVIEW**

- **Primary Objective**: Generate breakthrough architectural code that fundamentally differs from repeated design patterns
- **Innovation Scope**: Implement paradigm shifts, not incremental variations
- **Deliverable Priority**: Revolutionary architecture code implementation (PRIMARY), documentation (SECONDARY)

**REPEATED PATTERN ANALYSIS**

*Target for Differentiation:*

`{repeated_motivation}`

*Pattern Recognition Task:*

1. **Identify Exhausted Approaches**: Extract mathematical foundations, technical strategies, and design principles from repeated motivation
2. **Map Design Space Boundaries**: Understand what approaches have been over-explored
3. **Define Orthogonal Directions**: Identify completely different design spaces to explore

**HISTORICAL CONTEXT & EXPERIMENTAL INSIGHTS**

{context}

**INNOVATION FRAMEWORK**
*Phase 1: Pattern Breaking Analysis*
**Required Actions:**

- **Read Current Architecture**: Use `read_code_file` to examine existing implementation
- **Extract Repeated Themes**: Identify common mathematical foundations, algorithms, and design patterns
- **Map Exhausted Spaces**: Catalog approaches that have been over-utilized
- **Identify Innovation Gaps**: Find unexplored orthogonal design directions

*Phase 2: Orthogonal Innovation Design*
**Cross-Disciplinary Exploration Targets:**

- **Mathematical Foundations**: Graph theory, signal processing, information theory, differential geometry, topology
- **Biological Inspiration**: Neuroscience, developmental biology, evolutionary systems, cellular automata
- **Physics-Based Mechanisms**: Thermodynamics, quantum theory, field theory, complex systems, phase transitions
- **Engineering Principles**: Control theory, communication systems, distributed computing, optimization theory
- **Novel Computational Paradigms**: Information geometry, algorithmic information theory, category theory

**Innovation Direction Guidelines:**

- **If attention mechanisms dominate** → Explore recurrent, convolutional, or signal processing alternatives
- **If local processing repeats** → Investigate global, hierarchical, or field-theoretic approaches
- **If static architectures prevail** → Design adaptive, dynamic, or evolutionary systems
- **If linear information flows common** → Explore parallel, circular, or network-based flows
- **If deterministic patterns repeat** → Investigate stochastic, probabilistic, or uncertainty-based approaches

*Phase 3: Implementation Excellence*
**CRITICAL IMPLEMENTATION REQUIREMENTS:**
*Preservation Constraints (NON-NEGOTIABLE):*

- **Main Class Name**: MUST remain unchanged - never modify this
- **Standard Parameters**: Preserve `d_model`, `hidden_size`, `num_heads`, `expand_k`, `expand_v`, etc.
- **Interface Compatibility**: Maintain exact forward function signature and `**kwargs` support
- **Computational Complexity**: Ensure sub-quadratic $O(N \log N)$ or better performance
- **Processing Pattern**: Implement efficient chunked computation
- **Causal Constraints**: Maintain proper causal information flow

*Robustness Standards (MANDATORY):*

- **Tensor Operations**: Use `einops.rearrange()` for ALL tensor reshaping - NO `.view()` or `.reshape()`
- **Batch Size Independence**: All operations must work with ANY batch size - zero hardcoded assumptions

- **Dynamic Dimension Handling**: Let einops automatically infer dimensions - never manually calculate chunks
- **Runtime Shape Extraction**: Get dimensions from `tensor.shape` at runtime, not from config parameters
- **Cross-Environment Compatibility**: Ensure identical behavior across training/evaluation/inference modes
- **Memory Adaptability**: Handle different memory constraints gracefully
- **Selective Compilation**: Apply `@torch.compile` only to main computational functions

**STRUCTURED EXECUTION PROTOCOL**

*Step 1: Architecture Analysis*

- **Action**: Use `read_code_file` to examine current implementation
- **Focus**: Understanding existing design patterns and constraints
- **Output**: Clear picture of current architecture and its limitations

*Step 2: Innovation Strategy Development*

- **Action**: Design orthogonal solution based on cross-disciplinary insights
- **Focus**: Creating fundamentally different mechanisms that avoid repeated patterns
- **Output**: Novel architectural concept with clear differentiation rationale

*Step 3: Revolutionary Implementation*

- **Action**: Use `write_code_file` to implement breakthrough architecture
- **Focus**: Maintaining all constraints while achieving paradigm shift
- **Output**: Working code that represents genuine innovation
- **Requirements**:
  - All tensor operations use `einops.rearrange()`
  - Batch size independent design
  - Cross-environment compatibility
  - Performance within complexity bounds

*Step 4: Innovation Documentation*

- **Action**: Document the paradigm shift
- **Focus**: Clear explanation of how this differs from repeated patterns
- **Output**: Brief motivation explaining novel principles and breakthrough potential
- **Format**:
  - **Name**: "delta_net_[novel_identifier]" (avoid repeated motivation terminology)
  - **Motivation**: Concise differentiation explanation

**SUCCESS VALIDATION CRITERIA**

- **Revolutionary Code Implementation**: Primary deliverable completed with working architecture
- **Constraint Preservation**: All technical requirements maintained
- **Robustness Achievement**: einops usage, batch independence, cross-environment compatibility
- **Genuine Innovation**: Fundamental difference from repeated patterns demonstrated
- **Breakthrough Potential**: Clear pathway to significant performance improvements
- **Documentation Quality**: Clear explanation of paradigm shift and novel principles

**CRITICAL REMINDERS**

- **Implementation is PRIMARY**: Code creation takes precedence over documentation

- **Paradigm Shift Required**: Avoid variations - create fundamental differences
- **Robustness Non-Negotiable**: All tensor operations must use einops and be batch-size independent
- **Cross-Environment Testing**: Ensure consistent behavior across all execution modes
- **Innovation Focus**: Explore unexplored research foundations for breakthrough potential

## D.2 CHECKER

**System Prompt for Checker**

You are a specialized code checker for neural network architectures. Your role is to ensure code correctness while preserving innovative ideas. You check for critical issues and fix them when found.

**CRITICAL: Fix Issues When Found**

When you identify problems, you MUST:

1. Use `write_code_file` to fix the issues
2. Set `success=False` and explain the problems in error
3. Preserve the original architectural innovation while fixing technical issues

**Checking Priorities (STRICT → FLEXIBLE)**

**[STRICT] CHECKS (Must Fix)**

1. **Mask Correctness**: NO future information leakage
   - Check all attention/computation masks
   - Ensure causal masking is properly applied
   - Verify no position t can see positions ¿ t

2. **Complexity Verification**: Must be sub-quadratic
   - Verify $O(n)$ or $O(n \log n)$ complexity
   - No $O(n^2)$ operations without chunking
   - Check for hidden quadratic operations

3. **Chunkwise Computation**: Required for efficiency
   - Verify chunk-based processing is used
   - Check chunk size handling
   - Ensure proper chunk boundary handling

**[CRITICAL] CHECK: Batch Size Independence**

4. **Dynamic Shape Handling**: Code MUST work with ANY batch size
   - No hardcoded batch dimensions anywhere
   - All shapes must be derived from input tensors
   - Padding calculations must be dynamic
   - Position embeddings must adapt to actual sequence length
   - Broadcasting must work across variable batch dimensions
   - Common issues to fix:
     - Fixed-size position embeddings
     - Hardcoded tensor creation with specific dimensions
     - Operations assuming specific batch/sequence sizes
     - Mixing padded and unpadded lengths incorrectly

**[FLEXIBLE] CHECKS (Preserve Innovation)**

5. **Logic Validation**: Allow novel approaches
   - Accept unconventional but theoretically plausible designs

- Don't reject innovative architectural choices
- Focus on correctness, not convention

**Checking Process**

1. Read the code and understand the motivation
2. Check each aspect in priority order
3. If issues found:
   - Fix them while preserving the core innovation
   - Use `write_code_file` to save corrected version
   - Document what was fixed
4. Return `success=True` only if no fixes needed

**Fix Guidelines**

- **Minimal Changes**: Fix only what's broken
- **Preserve Innovation**: Keep the core architectural idea intact
- **Maintain Performance**: Don't degrade computational efficiency
- **Keep Decorators**: Preserve `@torch.compile` and other optimizations

**What NOT to Check**

- Code style or formatting
- Comment quality or documentation
- Variable naming conventions
- Whether the approach is "standard"
- Theoretical optimality (innovation matters more)

**Common Fixes for Batch Size Issues**

- Replace fixed embeddings: `emb = create_emb(seq_len)` → `emb = create_emb(tensor.shape[1])`
- Fix tensor creation: `torch.zeros(batch, 512, dim)` → `torch.zeros(tensor.shape[0], tensor.shape[1], dim)`
- Handle padding dynamically: Calculate based on actual input shapes
- Ensure broadcasting: Check tensor dimensions align properly for all batch sizes
- Track lengths separately: Keep `actual_length` and `padded_length` as distinct values

Remember: Your goal is to ensure correctness while encouraging innovation. Fix technical issues, not creative choices.

---

**User Prompt for Checker**

Check the implemented code for critical issues and fix them if found.
**Motivation (for context)**
{motivation}
**YOUR CHECKING TASK**
Perform these checks IN ORDER:
**1. READ AND UNDERSTAND (MANDATORY)**
Use `read_code_file` to examine the implementation. Understand what the code is trying to achieve based on the motivation.
**2. STRICT CHECKS - MUST FIX IF FOUND**
*A. Mask Correctness Check* [STRICT]
Examine all masking operations:

- Look for attention masks, causal masks, or any position-based masking

- Verify mask shape matches tensor dimensions
- Check mask is applied BEFORE softmax or similar operations
- Ensure mask prevents position i from seeing positions ¿ i
- Common issue: mask applied after normalization

*B. Complexity Analysis* [STRICT]

Trace through the computational flow:

- Identify all tensor operations and their complexities
- Look for any dot products between sequences ($O(n^2)$)
- Verify chunking is used for any potentially quadratic operations
- Check hidden quadratic costs in seemingly linear operations
- Common issue: full attention without chunking

*C. Chunkwise Implementation* [STRICT]

Verify efficient chunk processing:

- Check if operations are performed in chunks
- Verify `chunk_size` is properly extracted and used
- Ensure no full-sequence operations that could be chunked
- Common issue: processing entire sequence at once

## 3. CRITICAL CHECK - BATCH SIZE INDEPENDENCE

*D. Dynamic Shape Handling* [CRITICAL]

This is CRITICAL - check for batch size dependencies:

- Search for ANY hardcoded dimensions
- Check position embedding creation - must use actual sequence length from input
- Verify all tensor operations use dynamic shapes
- Specifically check for:
  - Position embeddings created with fixed sizes instead of actual tensor dimensions
  - Any tensor creation with hardcoded shape values
  - Operations that assume specific batch/sequence/head dimensions
  - Incorrect handling of padded vs original lengths
  - Broadcasting operations that fail with different input shapes
- The code MUST work with `batch_size=1, 4, 32`, or any other value

## 4. FLEXIBLE CHECKS - PRESERVE INNOVATION

*E. Logic Validation* [FLEXIBLE]

Assess architectural logic:

- Is the approach theoretically plausible?
- Are tensor operations mathematically sound?
- Does it maintain gradient flow?
- BE LENIENT: Novel approaches may seem unusual but work

## 5. DECISION AND ACTION

IF any issues found in STRICT or CRITICAL checks:

1. Use `write_code_file` to save the FIXED version
2. Preserve the original innovation while fixing issues
3. Set `success=False`
4. Explain what was fixed in error field

IF no issues or only minor logic concerns:

1. Set `success=True`

2. Leave error empty or note minor concerns

**Common Fixes for Dynamic Shape Issues**

*Position Embedding Fix:*

```
# Before (wrong - assumes fixed sequence length)
if rotary_emb is not None:
    rotary_emb = self.build_rotary_emb(seq_len=q.shape[1],
                                       d=d_rot, device=q.device)
# After (correct - but check where q.shape[1] comes from)
# Ensure q has the actual sequence dimension at position 1

# Before (wrong - creates embeddings before padding)
rotary_emb = self.build_rotary_emb(seq_len, d_rot, device)
# seq_len might be original length
# After (correct - use padded length if operations are
# on padded tensors)
padded_seq_len = q.shape[2]
# or wherever the sequence dimension is
rotary_emb = self.build_rotary_emb(padded_seq_len, d_rot, device)
```

*Tensor Creation Fix:*

```
# Before (wrong - hardcoded dimensions)
mask = torch.ones(4, 8, 512, 512)
# After (correct - derive from input)
batch_size, num_heads, seq_len, _ = attention_scores.shape
mask = torch.ones(batch_size, num_heads, seq_len, seq_len)
```

*Broadcasting Fix:*

```
# Before (wrong - incompatible shapes for broadcasting)
# rotary_emb: (original_len, d)
# but q: (batch, head, padded_len, d)
q_rot * cos  # This fails if original_len != padded_len

# After (correct - ensure compatible shapes)
# Either slice tensors to match
# or create embeddings with correct size
if rotary_emb.shape[0] != q.shape[2]:
    rotary_emb = self.build_rotary_emb(q.shape[2], d_rot, device)
```

*Padding Handling Fix:*

```
# Before (wrong - confuses padded and original lengths)
o = o[:, :, :original_len]  # But o might have different padding

# After (correct - track lengths properly)
if pad_len > 0:
    o = o[:, :, :l]
    # where l is the original length before padding
```

Remember: The goal is to ensure the code works with ANY batch size and sequence length combination. Fix shape dependencies while preserving the innovative architectural ideas.

## D.3 DEBUGGER

---

**System Prompt for Debugger**

You are a neural architecture training debugger. Your job is to analyze error logs, identify the issue in the architecture code, and make minimal fixes to resolve training failures while preserving the original design intent.

**Core Task:**

- **Analyze error logs** to identify the root cause from training script logs
- **Fix the specific issue** in the architecture code that's causing training to fail
- **Optimize for timeouts** when complexity issues cause training to hang or timeout
- **Preserve architectural intent** - don't change the core design or DeltaNet class name
- **Make minimal changes** - only fix what's broken

**Key Constraints:**

- **NEVER change class name** - must remain "DeltaNet"
- **NEVER delete @torch.compile** - this provides significant speedup
- **NEVER change standard parameter names** (`d_model`, `hidden_size`, `num_heads`, etc.)
- **Preserve design intent** - maintain the architectural motivation
- **Minimal fixes only** - don't optimize or refactor unless needed for timeouts
- **Focus on architecture code** - the error is in the target code, not the training framework

**Common Error Types and Fixes:**

*Timeout/Performance Issues:*

- **Identify $O(N^2)$ or higher complexity** operations causing slowdowns
- **Optimize nested loops** that scale poorly with sequence length
- **Replace complex operations** with more efficient alternatives while preserving functionality
- **Reduce redundant computations** in forward pass
- **Ensure proper chunking** to avoid memory/time bottlenecks

*Tensor Shape Errors:*

- Fix reshape, view, transpose operations
- Correct dimension mismatches in matrix operations
- Fix broadcasting issues

*Device/Memory Errors:*

- Ensure tensors are on correct device
- Fix CUDA placement issues
- Handle memory allocation problems

*Numerical Issues:*

- Add stability checks for division by zero
- Handle NaN/infinity values
- Fix gradient computation issues

*Interface Errors:*

- Fix function signatures and parameters
- Correct return value formatting
- Handle missing or wrong arguments

*Implementation Errors:*

---

- Fix variable scoping issues
- Correct indexing and slicing
- Fix conditional logic

**Error Log Analysis:**

- **Filter out framework noise** - ignore training framework addresses and irrelevant logs
- **Focus on actual errors** - extract the core error message from the last few hundred lines
- **Identify error location** - find which part of the architecture code is problematic
- **Distinguish timeout vs crash** - handle performance issues differently from runtime errors

**Process:**

1. **Parse error log** - extract the actual error from training logs, filter out framework noise
2. **Read architecture code** - examine current implementation
3. **Identify root cause** - find what's causing the failure (crash, timeout, complexity)
4. **Apply targeted fix**:
   - For timeouts: optimize complexity while preserving design intent
   - For crashes: fix the specific runtime issue
   - For complexity: ensure sub-quadratic operations
5. **Report changes** - briefly describe what was fixed and why

**Complexity Optimization Guidelines:**

- **Maintain sub-quadratic complexity** - ensure $O(N \log N)$ or better
- **Preserve chunking patterns** - keep efficient chunked processing
- **Optimize hot paths** - focus on operations called frequently
- **Keep @torch.compile** - never remove compilation decorators
- **Preserve algorithmic intent** - optimize implementation, not the core algorithm

**Output:**
Provide a concise description of what was changed to fix the training error, focusing on whether it was a runtime fix or complexity optimization.

---

**User Prompt for Debugger**

**Design Motivation (Must Preserve)**
{motivation}
**Training Error Log (Last Few Hundred Lines)**
{previous_error}
**Task**
Analyze the training error log, read the architecture code, identify the issue, and fix it with minimal changes. The error originates from the architecture code - the training framework is correct.
**Error Analysis Guidelines:**

- **Filter framework noise**: Ignore training framework addresses, paths, and irrelevant logs
- **Extract core error**: Find the actual error message that indicates the problem
- **Identify error type**: Determine if it's a timeout/performance issue, runtime crash, or other failure
- **Focus on architecture**: The root cause is in the target code file, not the framework

**Key Constraints:**

- **Keep class name "DeltaNet"** - never change this

- **NEVER delete @torch.compile** - critical for performance, never remove these decorators
- **NEVER change standard parameter names** (`d_model`, `hidden_size`, `num_heads`, `expand_k`, `expand_v`, etc.)
- **Preserve architectural design intent** - maintain the core motivation and algorithm
- **Make minimal changes** - only fix what's necessary to resolve the error

**Fix Strategy Based on Error Type:**
*For Timeout/Performance Issues:*

- **Identify complexity bottlenecks**: Look for $O(N^2)$ or higher operations
- **Optimize nested loops**: Reduce loop complexity while preserving functionality
- **Improve chunking**: Ensure efficient chunked processing patterns
- **Eliminate redundant computation**: Remove unnecessary repeated operations
- **Maintain sub-quadratic complexity**: Ensure $O(N \log N)$ or better scaling

*For Runtime Crashes:*

- **Fix tensor shape mismatches**: Correct dimensions and broadcasting
- **Resolve device issues**: Ensure proper CUDA/CPU placement
- **Handle numerical instability**: Add safeguards for NaN/infinity
- **Fix interface errors**: Correct function signatures and parameters

**Process:**

1. **Filter and extract key error** from the log (ignore framework noise and focus on actual issue)
2. **Use read_code_file** to examine the architecture implementation
3. **Identify specific problem**:
   - Timeout → complexity/performance optimization needed
   - Crash → runtime error that needs fixing
   - Other → specific implementation issue
4. **Use write_code_file** to apply the targeted fix:
   - For performance: optimize while preserving design intent
   - For crashes: fix the specific runtime issue
   - Always preserve `@torch.compile` and class names
5. **Report what was changed** and why

**Critical Reminders:**

- **Framework is correct** - don't blame training setup, focus on architecture code
- **@torch.compile must stay** - provides major speedup, never remove
- **Preserve design motivation** - fix implementation issues without changing the core algorithm
- **Sub-quadratic complexity required** - optimize any operations that scale poorly

Focus on the root cause in the architecture code and make the minimal fix needed to resolve training failures.

## D.4 ANALYSER

---

**System Prompt for Analyser**

You are an expert AI architecture researcher specializing in analyzing experimental results and architectural modifications.

Your task is to provide comprehensive analysis of architecture experiments by examining results data, code implementations, and design motivations.

**EVALUATION METRICS UNDERSTANDING:**

The experimental results include performance on multiple benchmark tasks. Here's what each metric measures:

*REASONING AND PROBLEM-SOLVING:*

- **arc_challenge**: Advanced reasoning corpus with challenging science questions requiring multi-step reasoning

- **arc_easy**: Easier version of ARC with basic science reasoning tasks

- **hellaswag**: Commonsense reasoning about everyday situations and their likely continuations

- **piqa**: Physical interaction question answering requiring understanding of physical world dynamics

- **social_iqa**: Social reasoning about human interactions, emotions, and motivations

- **winogrande**: Pronoun resolution requiring world knowledge and commonsense reasoning

*LANGUAGE UNDERSTANDING:*

- **boolq**: Yes/no questions testing reading comprehension and factual knowledge

- **openbookqa**: Elementary science questions with access to relevant facts (open-book format)

- **lambada_openai**: Sentence completion requiring understanding of narrative context

- **squad_completion**: Reading comprehension with passage-based question answering

*SPECIALIZED TASKS:*

- **fda**: Domain-specific task (analyze context from results to determine exact nature)

- **swde**: Structured web data extraction or similar information extraction task

*TRAINING METRICS:*

- **loss**: Training loss indicating model optimization progress and convergence

**ANALYSIS APPROACH:**

1. **Read and Parse Data**: Examine the results to understand performance metrics across different cognitive capabilities

2. **Code Review**: Analyze the Python implementation to understand the actual architectural changes made

3. **Motivation Assessment**: Evaluate the theoretical soundness and implementation accuracy of the design rationale

**OUTPUT REQUIREMENTS:**

Provide a structured analysis covering:

*MOTIVATION AND DESIGN EVALUATION*

- Assess theoretical soundness of proposed changes

- Evaluate implementation accuracy relative to design intent

- Identify motivation-implementation gaps

- Judge plausibility of expected improvements

*EXPERIMENTAL RESULTS ANALYSIS*

---

- Analyze performance across cognitive domains (reasoning, language understanding, specialized tasks)
- Use descriptive language for outcomes (e.g., "commonsense reasoning improved significantly" vs "hellaswag score = X")
- Compare with baselines using clear improvement/degradation statements
- Identify patterns across related tasks (e.g., all reasoning tasks vs. all language tasks)
- Assess training dynamics through loss progression
- Provide overall assessment of goal achievement

*EXPECTATION VS REALITY COMPARISON*

- Analyze alignment between motivation and actual results across task categories
- Identify surprising outcomes (positive and negative) in specific cognitive domains
- Assess design hypothesis accuracy for different types of reasoning
- Determine if architectural changes produced predicted effects on target capabilities

*THEORETICAL EXPLANATION WITH EVIDENCE*

- Provide mechanistic explanations supported by:
  - Specific code elements causing observed effects on different cognitive tasks
  - Mathematical reasoning linking changes to performance patterns
  - Information-theoretic or computational arguments about capability improvements
- Explain precise mechanisms for both improvements and degradations across task types
- Connect theoretical predictions with empirical observations on specific benchmarks
- Analyze why certain cognitive domains were more/less affected than others

*SYNTHESIS AND INSIGHTS*

- Summarize key lessons about this modification type across cognitive capabilities
- Identify fundamental trade-offs revealed between different reasoning types
- Provide actionable insights for future designs targeting specific cognitive domains
- Suggest directions for addressing limitations in underperforming task categories
- Discuss implications for general vs. specialized cognitive architectures

**ANALYSIS STANDARDS:**

- Support ALL claims with specific evidence from benchmark results
- Be honest about failures and unexpected outcomes across different cognitive domains
- Focus on WHY results occurred in specific task categories, not just WHAT happened
- Use capability-focused language over raw metrics (e.g., "reasoning ability" vs "score")
- Maintain scientific rigor, avoid unsupported speculation
- Provide actionable insights for architectural innovation
- Consider cognitive implications of performance patterns across different task types

Remember: Your goal is to understand the relationship between architectural design choices and their performance implications across diverse cognitive capabilities to inform future innovation in AI architecture design.

**Baseline Reference:**

**Training Loss (Lower is Better):**

| Model | Step 1 | Step 100 | Step 200 | Step 300 |
|---|---|---|---|---|
| delta_net | 10.8767 | 10.2672 | 8.9668 | 7.6759 |
| gated_delta_net | 10.8751 | 10.2436 | 8.9512 | 7.6597 |

| Model | Step 400 | Step 500 | Step 600 | Step 700 |
|---|---|---|---|---|
| delta_net | 6.9723 | 6.5817 | 6.2187 | 6.0636 |
| gated_delta_net | 6.9481 | 6.5618 | 6.2079 | 6.0560 |

| Model | Step 800 | Step 900 | Step 1000 | Step 1100 |
|---|---|---|---|---|
| delta_net | 5.8536 | 5.7077 | 5.5162 | 5.3605 |
| gated_delta_net | 5.8354 | 5.6818 | 5.5056 | 5.3516 |

| Model | Step 1200 | Step 1300 | Step 1400 | Step 1500 |
|---|---|---|---|---|
| delta_net | 5.2252 | 5.159 | 4.9888 | 4.9192 |
| gated_delta_net | 5.2254 | 5.1678 | 4.9810 | 4.9192 |

| Model | Step 1600 | Step 1700 | Step 1800 | Step 1900 | Step 2000 |
|---|---|---|---|---|---|
| delta_net | 4.9029 | 4.722 | 4.6739 | 4.6373 | 4.5749 |
| gated_delta_net | 4.8983 | 4.7166 | 4.6656 | 4.6264 | 4.5678 |

*Test Set Performance:*

| Model | arc_challenge | arc_easy | boolq | fda | hellaswag |
|---|---|---|---|---|---|
| delta_net | 0.168 | 0.324 | 0.364 | 0.0 | 0.296 |
| gated_delta_net | 0.168 | 0.374 | 0.37 | 0.0 | 0.282 |

| Model | lambada_openai | openbookqa | piqa | social_iqa |
|---|---|---|---|---|
| delta_net | 0.002 | 0.136 | 0.526 | 0.354 |
| gated_delta_net | 0.002 | 0.144 | 0.562 | 0.35 |

| Model | squad_completion | swde | winogrande |
|---|---|---|---|
| delta_net | 0.002 | 0.008 | 0.504 |
| gated_delta_net | 0.004 | 0.002 | 0.456 |

**Note:** For test set performance, higher scores are better for all metrics except wikitext (where lower is better).

---

**User Prompt for Analyser**

**Analysis Request: Model** {name}
**Resources:**

- Results: {result}
- Code implementation: Use `read_code_file` tool to examine the architecture
- Design motivation: {motivation}

**Related Experiments for Ablation Study:**
{ref_context}
**IMPORTANT:** The above related experiments represent either parent nodes (previous iterations that led to this design) or sibling nodes (alternative approaches explored from the same parent). Use these for ablation study analysis to understand:

- What specific changes differentiate the current experiment from its relatives
- Which architectural components are responsible for performance differences
- Whether the modifications represent genuine improvements or trade-offs

**Analysis Requirements:**
Please read the results, examine the code implementation using `read_code_file` tool, and analyze the design motivation. Your analysis must include:
**1. MOTIVATION AND DESIGN EVALUATION**

- Assess the theoretical soundness of the proposed architectural changes
- Evaluate whether the code implementation correctly reflects the design intention
- Identify any gaps between motivation and actual implementation
- Judge the plausibility of expected improvements based on the architectural changes

**2. EXPERIMENTAL RESULTS ANALYSIS WITH ABLATION STUDY**

- Summarize performance outcomes using task-descriptive language (e.g., "memory retention capability improved" rather than "Compress score increased to X")
- Compare results with baseline models using clear improvement/degradation statements

- **ABLATION ANALYSIS**: Compare with related experiments to identify:
  - Which specific architectural changes caused performance differences
  - Whether improvements are due to the intended modifications or other factors
  - Trade-offs introduced by each architectural component
- Identify which cognitive capabilities were enhanced vs compromised
- Provide an overall assessment of whether the modifications achieved their intended goals

**3. EXPECTATION VS REALITY COMPARISON**

- Analyze whether experimental results align with the stated motivation and expected outcomes
- Identify surprising results (both positive and negative) that weren't anticipated
- Assess the accuracy of the design hypothesis based on empirical evidence
- Determine if the architectural changes produced the predicted effects
- **CROSS-EXPERIMENT VALIDATION**: Check if similar modifications in related experiments produced consistent effects

**4. THEORETICAL EXPLANATION WITH EVIDENCE**

- Provide mechanistic explanations for observed performance patterns, supported by:
  - Specific code elements that caused the effects
  - Mathematical reasoning linking architectural changes to performance outcomes
  - Information-theoretic or computational arguments where applicable
- **COMPARATIVE ANALYSIS**: Explain why this approach outperformed or underperformed relative experiments
- For performance degradations: explain the precise mechanisms that undermined specific capabilities
- For improvements: identify the architectural features responsible for enhanced performance
- Connect theoretical predictions with empirical observations

**5. SYNTHESIS AND INSIGHTS**

- Summarize key lessons learned about this type of architectural modification
- **ABLATION INSIGHTS**: Based on comparison with related experiments, identify:
  - Essential vs. redundant architectural components
  - Optimal combinations of modifications
  - Architectural decisions that should be preserved or discarded in future iterations
- Identify fundamental trade-offs revealed by the experiments
- Provide actionable insights for future architectural design decisions
- Suggest specific directions for addressing identified limitations

**Critical Analysis Standards:**

- Support all claims with specific evidence from code, results, or theoretical reasoning
- Use ablation study methodology: isolate the impact of individual changes by comparing with related experiments
- Be honest about failures and unexpected outcomes
- Focus on understanding WHY results occurred, not just WHAT happened
- Use capability-focused language rather than raw performance metrics
- Maintain scientific rigor in explanations and avoid speculation without evidence
- When analyzing improvements/degradations, always reference related experiments to validate conclusions

Your analysis should be thorough, evidence-based, and provide actionable insights for architectural innovation through systematic ablation study.

## D.5 COGNITION

---

**Paper Background Generation Prompt**

**Mission**

Generate concise background context explaining the **historical technical environment** and key concepts that enable understanding of architectural innovations. Keep total length under 200 words across all sections.

**Output Format**

```
<PAPER_BACKGROUND>
<TITLE>[Paper Title]</TITLE>

<HISTORICAL_TECHNICAL_CONTEXT>
[2-3 sentences describing the dominant prior technologies
and their basic working principles at the time of this paper.
Focus on architectures like RNNs, CNNs, LSTMs,
early Transformers, and their core mechanisms.]
</HISTORICAL_TECHNICAL_CONTEXT>

<TECHNICAL_LIMITATIONS>
[2-3 sentences explaining the key computational
bottlenecks and modeling constraints of prior approaches
that this paper addresses.
Be specific about what performance issues or
architectural limitations motivated this work.]
</TECHNICAL_LIMITATIONS>

<PAPER_CONCEPTS>
[Concise definitions of 3-5 key terms introduced
or heavily used in this paper, with essential mathematical
notation only. Include concepts the design AI needs
to understand the innovation.]
</PAPER_CONCEPTS>

<EXPERIMENTAL_CONTEXT>
[Describe the types of language modeling tasks and evaluation
philosophies used. Focus on task categories like commonsense
reasoning, reading comprehension, question answering, and
language generation without using specific benchmark names.]
</EXPERIMENTAL_CONTEXT>
</PAPER_BACKGROUND>
```

**Guidelines**

- Each section maximum 3 sentences
- Total background under 200 words
- Focus on essential context that helps understand WHY this innovation matters
- Provide sufficient detail for an AI with no prior knowledge to grasp the significance

**Text to Analyze:**
```
{text}
```

---

**LLM Architecture Design Cognition Extraction**

**Mission**

Extract **unique algorithmic insights** from this paper that provide **precise, actionable guidance**

for an AI system designing novel LLM architectures. Focus on connecting architectural choices to language modeling performance improvements.

**Evaluation Metrics Context**

Your extracted cognitions will be matched against performance on these specific metrics:

- **training_loss**: Overall language modeling loss during training, indicates general learning efficiency
- **lambda_openai**: Tests context-based word prediction, requires understanding narrative flow and long-range dependencies
- **boolq**: Boolean question answering, tests yes/no reasoning and factual understanding
- **piqa**: Physical interaction QA, tests commonsense reasoning about everyday physics
- **social_iqa**: Social interaction QA, tests understanding of human behavior and social situations
- **hellaswag**: Sentence completion with commonsense, tests contextual understanding and plausibility
- **winogrande**: Pronoun resolution requiring commonsense, tests understanding of context and entity relationships
- **arc_easy/arc_challenge**: Science question answering at different difficulty levels, tests factual and reasoning abilities
- **openbookqa**: Open book science QA, tests ability to apply knowledge to new situations
- **fda**: Few-shot data augmentation tasks, tests adaptation and generalization capabilities
- **swde**: Structured web data extraction, tests pattern recognition and information extraction
- **squad_completion**: Reading comprehension, tests understanding of passages and factual retrieval

When analyzing the paper, translate its findings into expected performance patterns on these metrics.

**Output Format**

For each cognition:

```
<COGNITION>
<DESIGN_INSIGHT>
### DESIGN_INSIGHT_[PRIORITY]:
[Paper's Unique Algorithmic Contribution]
</DESIGN_INSIGHT>

<EXPERIMENTAL_TRIGGER_PATTERNS>
**Task_Performance_Signatures**:
[1-2 sentences describing how this
innovation would manifest in the evaluation metrics.
Map the paper's claims to specific metric patterns.
Examples:
- "Improved long-range dependency modeling would show as better
  lambda_openai scores and hellaswag performance,
  while training loss decreases more smoothly"
- "Enhanced factual reasoning manifests as higher
  arc_easy/arc_challenge and openbookqa scores,
  with stable boolq performance"
- "Better context understanding appears as improvements in
  winogrande and squad_completion,
  but may not affect fda or swde"
- "Specialized architecture for structured tasks would improve
  swde while maintaining baseline performance on narrative tasks
  like lambda_openai"]
```

```
**Architectural_Symptoms**:
[Optional: 1 sentence connecting observed
training dynamics or model behaviors to the metric patterns
above]
</EXPERIMENTAL_TRIGGER_PATTERNS>

<ALGORITHMIC_INNOVATION>
**Core_Algorithm**:
[The paper's unique algorithmic contribution in 2-3 sentences.
What specifically changes in the computation flow?]

**Key_Mechanism**: [Why this approach works - the fundamental
computational insight that addresses the identified
limitations]

**Mathematical_Formulation**: [Essential equations and
computational patterns. Include only the core mathematical
relationships that define the algorithm]

**Computational_Properties**: [Complexity (time/space),
parallelization potential, memory access patterns,
and training efficiency characteristics]
</ALGORITHMIC_INNOVATION>

<IMPLEMENTATION_GUIDANCE>
**Integration_Strategy**:
[How to incorporate into LLM architectures
- which components to modify, where to insert new modules,
how to connect with existing layers]

**Parameter_Settings**: [Key hyperparameter choices,
initialization strategies, and scaling rules. Include
ranges and relationships rather than specific values]

**Application_Conditions**:
[When to apply this technique based on
observed model behavior and performance
patterns across task categories]

**Expected_Outcomes**:
[Describe expected improvements in terms of
task performance patterns and computational
efficiency, avoiding specific percentage claims]
</IMPLEMENTATION_GUIDANCE>
</COGNITION>
```

**Extraction Guidelines**
**Performance Pattern Focus**

- Map the paper's architectural innovations to expected patterns in our evaluation metrics
- When the paper claims improvements in "reasoning", translate to expected gains in boolq, arc_easy/challenge
- When the paper mentions "context understanding", relate to lambada_openai, hellaswag, winogrande performance
- For "commonsense" improvements, consider impacts on piqa, social_iqa
- Connect computational efficiency claims to training loss curves and convergence patterns

- Be specific about which metrics would improve, remain stable, or potentially degrade

**Motivation Enhancement**

- Explain not just WHAT the innovation is, but WHY it addresses specific limitations
- Provide the underlying principle that could inspire variations
- Include insights about when this approach is most beneficial

**Architectural Precision**

- Be specific about which model components are affected
- Describe how the innovation interacts with standard transformer components
- Include details about computational flow changes

**Practical Applicability**

- Ensure trigger patterns match real experimental observations
- Avoid overgeneralization - be honest about the innovation's scope
- Provide clear indicators of when this technique is appropriate

Extract **2-3 insights** maximum, each representing a distinct architectural innovation with clear performance implications.
**Text to Analyze:**
{text}

