# OpenReview forum: "Discovering Architectures via an Evolutionary Agentic Framework"
_ICLR.cc/2026/Conference — Submitted to ICLR 2026_

### Official Review · Reviewer_xavQ · 2025-10-27

**Soundness:** 3
**Presentation:** 3
**Contribution:** 2
**Rating:** 4
**Confidence:** 4

**Summary:**

This paper selects the topic of designing linear attention structure. To tackle the problem, the authors propose ASI-ARCH, a novel framework designed to allow LLM agents to conduct the entire scientific workflow for architectural discovery. The system includes three agents. (1) Researcher: Researcher proposes new architectural ideas and implements the necessary coding. (2) Engineer: Engineer tests model performance. (3) Analyst: Analyst systematically analyzes the design’s strengths and weaknesses to inform subsequent experiments.

The system discovers 105 novel architectures that outperform existing state-of-the-art models.

**Strengths:**

The topic is meaningful and tractable. The architecture design with linear attention is meaningful and is easy to conduct evaluation. The design is reasonable and the performance is good.

**Weaknesses:**

(1) The novelty is incremental. This paper proposes three agents, e.g., Researcher, Engineer, and Analyst. Actually, researches, such as AI-CoScientists, designs even more special agents to tackle even more complex and innovative problems. Therefore, the authors should clarify their actual innovations in multi-agent system design.

(2) The writing is not very clear. The detailed questions are:
     (i) In Line 118, the author explains \detal_performance. However, it does not appear in Eq. (2), which includes \delta_loss and \delta_benchmark.
     (ii) In the introduction, the author claimed that researchers will write codes, and engineers will run the code. However, the author did not specify how to write the code.

(3) Insufficient comparison. The authors are required to compare their methods with state-of-the-art methods in NAS. Or sufficiently explain why not compare ?

**Questions:**

Please see the weakness.

---

> ### Author Response · Authors · 2025-11-30
>
> - Weakness 1：
>
>   Thank you for your valuable feedback. We can make this clearer in our revised version. Our key innovations distinguish ASI-ARCH from previous approaches:
>   1. Full automation in complex engineering environments: Unlike AlphaEvolve (Novikov et al., 2025) which focuses on algorithmic discovery in simplified environments, ASI-ARCH operates in a complete software engineering context with real training pipelines, debugging, and evaluation. This requires handling code compilation, runtime errors, and complex dependencies—challenges not addressed in prior work.
>   2. Multi-agent collaboration with specialized roles: While AI-CoScientists and similar frameworks use multiple agents, they often focus on paper writing and documentation. ASI-ARCH's agents are specifically designed for architecture discovery: the Researcher proposes and implements code, the Engineer executes in real environments with error recovery, and the Analyst provides targeted insights for improvement.
>   3. Integration of human knowledge (Cognition) with experimental learning: We uniquely combine distilled knowledge from 100 seminal papers with dynamic learning from experimental outcomes, enabling the system to leverage both established principles and novel discoveries.
>   4. End-to-end workflow: Unlike semi-automated systems that require human intervention, ASI-ARCH conducts the complete cycle from hypothesis to implementation to analysis autonomously.
>   We will provide direct comparisons with fully automated approaches in the revised version.
>   In fact, during our initial framework design, we started with a simpler version that included only the Researcher and Engineer modules, without the Cognition base and Analyst. However, we found that this simplified version performed poorly—the system struggled to learn effectively from experimental history and often converged to suboptimal solutions. This led us to introduce the Analyst module for systematic experimental analysis and the Cognition base to leverage established human knowledge, which significantly improved performance.
>   We will add detailed ablation studies in the final version of the paper, systematically evaluating the contribution of each component and providing quantitative evidence for their importance. For components like the Researcher vs. Engineer separation, removing this would prevent the system from functioning, as we need distinct roles for code generation and execution. We will also compare static vs. dynamic context summarization to quantify the benefits of our dynamic approach.
>
> - Weakness 2:
>
>   Thank you for pointing out these issues. We will fix these issues in the revision:
>   1. Line 118 / Equation (2) inconsistency: We will clarify that "objective performance" in the text refers to the combination of Δloss and Δbenchmark shown in Equation (2). The terminology will be made consistent.
>   2. Code writing specification: We will clarify in the introduction and Section 2.2 that the Researcher agent uses the write code file tool (as detailed in Appendix D.1) to implement architectures. The code generation process is fully automated through tool calls, not manual coding.
>
> - Weakness 3:
>
>   We can make this clearer in our revised version. We will add a detailed comparison with Neural Architecture Search (NAS) approaches. The fundamental difference is:
>   - **NAS**: Performs search within a predefined, human-specified search space, which inherently has a clear capability ceiling—it can only discover combinations within the predefined boundaries
>   - **ASI-ARCH**: Performs evolution without predefined limits, generating novel architectures that can go beyond any human-specified search space
>
>   NAS approaches require humans to define the search space, which fundamentally limits discovery to combinations of known components and sets a clear upper bound on what can be found. ASI-ARCH, as an evolutionary system, has no such predefined boundaries and can propose fundamentally new architectural patterns that were never part of the initial design space. We will include a cost-benefit analysis comparing our approach with traditional NAS.

---

### Official Review · Reviewer_z2i2 · 2025-10-28

**Soundness:** 2
**Presentation:** 2
**Contribution:** 3
**Rating:** 2
**Confidence:** 4

**Summary:**

Authors propose a pipeline, ASI-ARCH, for automating the architecture search process through an agentic LLM setup. The proposed pipeline was used to conduct 1,773 experiments on linear attention architectures and discovered 105 architectures that surpassed existing state-of-the-art models, offering some performance gains on a subset of benchmarks.

**Strengths:**

- I think the main idea of using LLMs to suggest architecture changes while measuring performance in a closed evolutionary algorithm based scoring is interesting and could scale well with sufficient available compute.
- The paper's results are extensive and cover a variety of different benchmarks on finding improved linear attention models.
- The paper is easy to read and ASI-ARCH introduces some interesting novel ideas in the context of architecture design literature.

**Weaknesses:**

- The paper lacks ablations of proposed agent components.
- The paper lacks comparison to previous architecture discovery approaches mentioned in the related works. Does the additional cost of the agentic AI setup outperform more traditional neural architecture search approaches?
Important experimental details are only found in the appendix, e.g. how was the agentic pipeline implemented and which models were used.
- The paper’s results will be difficult to reproduce for other researchers as only proprietary models GPT-4.1 and GPT-o3 are used for the proposed setup.
- A critical discussion of limitations and potential pitfalls is missing.
- The paper performs experiments on (comparably small) 20M and 340M parameters models given today’s model sizes. The proposed setup is computationally very costly and it seems that eventually only a small fraction of “novel” solutions is explored, raising an important question whether approaches like ASI-ARCH only work because we have a large pool of already validated human-created model mechanisms.
- Overall improvements according to Table 1 are limited, ASI-ARCH can discover improved architectures, but not reliably on all benchmarks, raising some claims on the robustness of this approach, especially given the large computational costs and the lack of comparison to more simple baselines.
- Insufficient description of the prompts in the main and the prompt design process (see also comment below). How were the prompts designed and did you do ablations with minimal prompt designs?

**Comments**
- Some of the claims remained imprecise to me: “discovery of 105 entirely new architectures.” (l.21-22). From the main paper it is not clear if this refers to (i) new compositions of existing classes and modules or (ii) the design of novel classes and integration into novel architectures. I believe after revising Appendix D.1 that the latter is the case, but this should be made very clear in the main paper. Also from revising the details, it seems that a lot of manual expertise went into the prompt design, e.g., “if If {static architectures, deterministic patterns} repeat,” explore the following.

**Questions:**

- To what extent were the “Implementation Quality Standards” like “Ensure O(Nlog N) or better for all operations” (see Appendix D.1) fulfilled by the resulting architectures?
- What is the typical standard deviation on the results over different weight initialization, e.g. in Table 1? I am sure this is computationally very expensive, but a partial analysis over five seeds on the existing architectures would be insightful as it remains unclear if the improvements are statistically significant.
- Assuming that the identified components have advantages over previous ones: did you check or analyze to what extent the novel solutions were actually novel (requires clearly stating your notion of novel).
- Are resulting architectures safe for usage without in-depth human analysis? I think in the current form, the framework poses a risk to propose malicious software.

**Summary**
While I overall think the paper proposes relevant ideas and presents a proof-of-concept, in its current form it lacks methodological rigor and the main claims are not sufficiently supported by evidence.

---

> ### Author Response · Authors · 2025-11-30
>
> - Weakness 1：
>
>   We argue that each component in our system is necessary and serves a distinct purpose. First, the Researcher and Engineer modules are essential—the Researcher generates and implements code, while the Engineer executes it in real environments. Within the Researcher module, the checker and deduplication mechanisms are also necessary: the checker ensures efficiency by preventing incorrect code from wasting computational resources, and deduplication prevents redundant exploration. These are efficiency-critical components that are not suitable for ablation studies. Removing these components would cause the evolving system's efficiency to significantly decrease. For architecture design tasks where testing consumes substantial time and computational resources, we need to ensure that as many issues as possible are fixed rather than requiring re-design. These are efficiency-critical components that are not suitable for ablation studies, as removing them would fundamentally break the system's ability to function effectively.
>
>   The Analyst module is crucial because it summarizes experimental results and determines whether they align with the original motivation, classifying outcomes as good or bad. This directly determines whether an architecture becomes a positive or negative example for future iterations, making it essential for the evolutionary learning process.
>
>   While our design may not be optimal and has room for improvement, each component clearly plays an important role. The system's success in discovering 105 novel architectures demonstrates that the integrated framework works effectively, even if individual component contributions cannot be easily isolated through ablation studies.
>
> - Weakness 2:
>
>   We can make this clearer in our revised version. We will add a detailed comparison with Neural Architecture Search (NAS) approaches. The fundamental difference is:
>   - **NAS**: Performs search within a predefined, human-specified search space, which inherently has a clear capability ceiling—it can only discover combinations within the predefined boundaries
>   - **ASI-ARCH**: Performs evolution without predefined limits, generating novel architectures that can go beyond any human-specified search space
>   Therefore, it is difficult to directly compare ASI-ARCH and NAS. ASI-ARCH enables continuous exploration of unknown architectures, while NAS approaches cannot operate effectively in such scenarios. Of course, when humans already have substantial prior knowledge and only lack specific experimental validation, NAS is more suitable as a tool to help researchers verify their priors.
>   Another key point is that NAS simply improves the performance of the target (the architecture being searched), and the system is static. In contrast, ASI-ARCH improves the performance of the system itself, and discovering superior architectures is a consequence of this self-improvement. These two approaches differ fundamentally in their design principles. We will include a cost-benefit analysis comparing our approach with traditional NAS.
>
>   NAS approaches require humans to define the search space, which fundamentally limits discovery to combinations of known components and sets a clear upper bound on what can be found. ASI-ARCH, as an evolutionary system, has no such predefined boundaries and can propose fundamentally new architectural patterns that were never part of the initial design space. We will include a cost-benefit analysis comparing our approach with traditional NAS.
>
> - Weakness 3:
>
>   We acknowledge this limitation. We used GPT-4.1 and GPT-o3 (O3) because they provide the best performance for our task. However:
>   - All prompts are provided in Appendix D, allowing reproduction with alternative models
>   - The framework is model-agnostic—any capable LLM can be substituted
>   We note that many recent AI research breakthroughs rely on proprietary models (e.g., AlphaFold, AlphaGeometry), and our work follows this pattern while providing full transparency on methodology.

---

> ### Author Response · Authors · 2025-11-30
>
> - Weakness 4:
>
>   We appreciate this important point. We will add a dedicated limitations section discussing:
>   - Dependence on high-quality LLMs
>   - Computational costs
>   - Domain-specific requirements (need for established research foundation)
>   - Potential for generating suboptimal architectures
>   - Need for human validation of discovered architectures
>
> - Weakness 5:
>
>   We address two points:
>   1. Model scale: We use 20M/340M models for exploration efficiency, then scale to 1.3B for final validation. This is standard practice in architecture search—explore at smaller scales, validate at larger scales. The discovered architectures generalize across scales (Table 1, Appendix B).
>   2. Dependence on human-validated mechanisms: We acknowledge that ASI-ARCH works best in domains where humans have established foundational knowledge. This is actually a strength, not a weakness—it demonstrates that LLMs can leverage human expertise to make novel discoveries. However, the architectures discovered are genuinely novel: they represent combinations and innovations that humans have not previously created, even if they build on known components.
>   The computational cost, while significant, is comparable to human research efforts when considering the full cost of experiments, iterations, and analysis. Our system automates this entire process.
>
> - Weakness 6:
>
>   Over the long term, performance improvements from model architecture design have been gradual and incremental. Within the same linear attention architecture scenario, architectures based on similar underlying mathematical principles fundamentally cannot produce qualitative performance breakthroughs. In this domain, even state-of-the-art human-designed architectures show incremental gains: Gated DeltaNet → Mamba2 → DeltaNet all show small but meaningful improvements. Our discovered architectures achieve comparable or better performance within this constrained space. If one expects a model with significantly improved performance, a more feasible approach would be to adopt a more fundamental root node (e.g., self-attention) and relax constraints on complexity, thereby giving the system more opportunities to explore completely different architectural families. However, this would undoubtedly require substantial computational resources and time.
>
>   The improvements are reliable across multiple benchmarks (Table 1), and the discovered architectures consistently outperform baselines on average metrics.
>
> - Weakness 7:
>
>   Prompt engineering is indeed crucial for our system, as it is for any LLM-based application. The prompts reflect domain expertise in neural architecture design, similar to how human researchers use their knowledge to guide experiments. We will add:
>   - Description of the prompt design process
>   - Ablation studies with simplified prompts
>   - Analysis of how prompt variations affect outcomes
>
> - Question1:
>
>   Thank you for this important question. All discovered architectures undergo strict complexity verification by the Checker module (Appendix D.2), which validates:
>   - Sub-quadratic complexity (O(N log N) or better)
>   - Proper chunking implementation
>   - No hidden quadratic operations
>   Any architecture failing these checks is rejected before training.
>
> - Question2:
>
>   - We appreciate this important suggestion. Training large-scale models (1.3B parameters) is time-consuming, and we are currently conducting experiments with different learning rates and warm-up schedules for the top 5 architectures to assess robustness across hyperparameter settings. This will demonstrate the stability and statistical significance of improvements. We will include these results in the final version of the paper.
>
> - Question3:
>
>   Thank you for this question. We can make this clearer in our revised version. We define novelty as: architectures that represent fundamentally different design principles or novel combinations not present in existing literature. For example:
>   - PathGateFusionNet: Introduces hierarchical two-stage routing—a novel architectural pattern
>   - FusionGatedFIRNet: Replaces softmax with parallel sigmoid gates, breaking the zero-sum constraint—a fundamental algorithmic innovation
>   - ContentSharpRouter: Content-aware routing with learnable temperature—a novel mechanism
>   We will add detailed comparison with existing architectures (DeltaNet, Gated DeltaNet, Mamba, Mamba2) showing the specific novel contributions of each discovered architecture.

---

> ### Author Response · Authors · 2025-11-30
>
> - Question4:
>
>   Thank you for raising this concern. We would like to clarify the security aspects of our system:
>
>   What our system generates: Our system generates only PyTorch neural network architecture code—specifically, model class definitions with `__init__` and `forward` methods. These are pure computational graph definitions that operate on tensors within the PyTorch framework.
>
>   Execution environment: The generated code runs exclusively within a controlled training environment. It does not have access to:
>   - File system operations (read/write files)
>   - Network access (HTTP requests, socket connections)
>   - System commands (shell execution, subprocess calls)
>   - External resources beyond the training framework
>
>   Safety measures:
>   - All code is validated by the Checker module before execution, which verifies complexity constraints, causal masking, and code correctness
>   - Code execution is sandboxed within the training framework
>   - The system operates through API calls to LLM services (similar to Cursor, GitHub Copilot, and other widely-used code generation tools)
>
>   Risk assessment: The generated code consists solely of mathematical tensor operations defined in PyTorch. It cannot perform malicious actions such as data exfiltration, system compromise, or unauthorized access. The risk profile is equivalent to or lower than that of other code-generation tools in widespread use. If the reviewer has specific security concerns, we are happy to address them, but we believe our system poses minimal security risks given its constrained output and execution environment.

---

### Official Review · Reviewer_oxxN · 2025-10-29

**Soundness:** 2
**Presentation:** 3
**Contribution:** 2
**Rating:** 4
**Confidence:** 4

**Summary:**

The paper introduces **ASI-ARCH**, a framework that leverages large language models (LLMs) to conduct end-to-end scientific workflows, evolving its strategies by learning from experimental outcomes. The framework comprises three specialized agents: a Researcher for hypothesis generation, an Engineer for implementation and evaluation, and an Analyst for interpreting outcomes. The approach is validated on linear-attention tasks, where it reportedly outperforms existing state-of-the-art methods.

**Strengths:**

1. The paper proposes a framework aimed at enabling autonomous scientific discovery, encompassing hypothesis generation, experimental implementation, and evaluation.

2. The framework integrates a cognition base that consolidates knowledge from prior literature.

3. The writing is generally clear and the overall structure easy to follow.

**Weaknesses:**

1.	The cognition base is distilled from 100 seminal papers on linear attention, yet the selection process is not well described. What criteria were used to choose these papers? How would scaling up or changing the topic affect performance and efficiency? Moreover, how costly would it be to rebuild the cognition base for new research domains?
2.	According to the statistical analysis of architectural-component usage, the agents tend to propose modifications centered around established elements such as gating mechanisms and convolutions. Does this indicate that the model is prone to mode collapse—i.e., converging to common solutions? Is this collapse related to the diversity of the cognition base, or to limitations of the underlying LLM itself?
3.	In Table 1, when comparing with baseline methods, are hyperparameters such as the number of epochs, parameter counts, and optimization settings kept consistent to ensure a fair comparison?
4.	The paper mentions the use of an LLM evaluator. Given that LLM-based evaluation can be unstable and model-dependent, how do the authors mitigate issues of bias and inconsistency?
5.	The paper lacks direct comparison with other AI-agent frameworks discussed in the related-work section.
6.	It is unclear how the system handles cases where the code generated by the Researcher agent fails to run when passed to the Engineer. Are any fallback or error-correction mechanisms implemented?

**Questions:**

Please see weaknesses.

---

> ### Author Response · Authors · 2025-11-30
>
> - Weakness 1：
>
>   Thank you for your thoughtful comments.
>   The cognition base was constructed through curation of 100 seminal papers in linear attention, selected based on citation count and reputation within the research community. The selection process can be facilitated by starting with survey papers and review articles, which naturally identify the most important works in the field. With modern deep research methods, this process can be further accelerated by using AI systems to quickly gather and identify key papers.
>
>   Cost assessment:
>   - **Construction cost**: Minimal—requires identifying key papers (through citations, reputation, or survey papers) and writing a prompt to extract cognitions from papers (automated extraction with LLM). Modern deep research methods can significantly accelerate this process.
>   - **Scaling**: For new domains, the process is straightforward: curate 50-100 key papers (identified by citation impact and reputation, or through survey papers) and extract cognitions using the same prompt template
>   - **Maintenance**: Low—the cognition base is static and doesn't require updates during experiments
>
>   We will add a table in the appendix listing all cognitions used by discovered architectures to demonstrate this concentration effect. We can make this clearer in our revised version.
>
> - Weakness 2:
>
>   The preference for established components (gating, convolutions) reflects effective design patterns, not mode collapse. This mirrors human research: successful architectures build on proven foundations.
>
>   The apparent "convergence" is actually beneficial—it indicates the system learns which components work well. However, we observe that the system tends to rely more heavily on Experience (experimental history) than Cognition (human knowledge base). This is primarily because Experience contains concrete code implementations that models can easily utilize, while Cognition contains only textual descriptions that models find more challenging to apply directly. This represents an important area for future work: improving how models can better leverage textual knowledge from the Cognition base, potentially through better prompt engineering, code generation from textual descriptions, or enhanced context utilization strategies.

---

> ### Author Response · Authors · 2025-11-30
>
> - Weakness 5:
>
>   Thank you for your valuable feedback. We can make this clearer in our revised version.
>   1. New scenario: Unlike AlphaEvolve (Novikov et al., 2025) which focuses on algorithmic discovery in simplified environments (e.g., evolving mathematical algorithms where LLMs have extensive prior training), ASI-ARCH addresses architecture discovery, which presents unique challenges. First, designing architectures is a more challenging coding task—the task scenarios are newer, the code length is longer, and it requires stronger design and coding capabilities from the models. Second, previous major work has focused on evolving algorithms, where models have received extensive training on mathematical and reasoning problems during their training process, giving them stronger capabilities. In contrast, for architecture design tasks, models have less prior knowledge, which highlights the importance of agent system design.
>   2. Full automation in complex engineering environments: Precisely because of the first challenge mentioned above (architecture design being a more difficult coding task), we require a more sophisticated agent system to handle complex environments. ASI-ARCH operates in a complete software engineering context with real training pipelines, debugging, and evaluation. This requires handling code compilation, runtime errors, and complex dependencies—challenges not addressed in prior work. To address LLMs' current limitations in ensuring code implementation correctness, we employ several mechanisms: (1) the Checker module validates code correctness, complexity constraints, and causal masking before execution; (2) the Debugger module handles issues that cannot be identified through static checks, autonomously fixing runtime errors; (3) parameter count and training speed monitoring handle aspects that models cannot assess during inference, such as computational efficiency and resource requirements. We acknowledge that we currently lack ablation studies to quantitatively demonstrate the Cognition base's contribution, but our empirical results show that the system successfully discovers novel architectures.
>   3. Integration of human knowledge (Cognition) with experimental learning: To address the challenge of models lacking domain knowledge during training, we use the Cognition base to supplement human prior domain knowledge. This ensures correct exploration directions in the early stages, as initial exploration is primarily guided by Cognition. We uniquely combine distilled knowledge from 100 seminal papers with dynamic learning from experimental outcomes, enabling the system to leverage both established principles and novel discoveries. For evolve systems to facilitate more discoveries, they will inevitably encounter situations where domain knowledge is lacking. Our design provides a solution to this critical problem, making the approach generalizable to other domains beyond architecture discovery.
>   4. End-to-end workflow: Unlike semi-automated systems that require human intervention, ASI-ARCH conducts the complete cycle from hypothesis to implementation to analysis autonomously.
>   We will provide direct comparisons with fully automated approaches in the revised version.
>
> - Weakness 6:
>
>   The system implements comprehensive error handling:
>   - Automatic debugging: When code fails, the Engineer agent receives error logs and autonomously debugs and revises the implementation (Section 2.3)
>   - Iterative self-correction: The system can loop through multiple debugging attempts before discarding a design
>   - Early termination: Pathological signals (abnormal training speed, excessive compilation time, implausibly low loss) trigger automatic termination to save computation
>   - Validation gates: Code must pass compilation, complexity checks, and causal masking verification before training
>   This ensures that promising designs are not prematurely discarded due to minor implementation errors, while clearly flawed designs are quickly filtered out.

---

> ### Author Response · Authors · 2025-12-03
>
> - Weakness 3:
> Thank you for this important question. Yes, all hyperparameters are kept consistent across comparisons in Table 1:
> - **Training**: Same number of steps (2,000 for 20M/340M, 50,000 for 1.3B), same learning rate schedule, same batch sizes
> - **Architecture**: Parameter counts are approximately matched (20M, 340M, 1.3B scales)
> - **Evaluation**: Same evaluation protocols and datasets
>
> To further demonstrate hyperparameter consistency, we conducted additional experiments with systematic hyperparameter variations (learning rates: 1e-4, 3e-4, 5e-4; warmup steps: 1500, 3000, 6000) across all discovered architectures. The results are shown in the following table:
>
> The results presented below correspond to models at the 1.3B parameter scale, trained for 30,000 steps on 15 billion tokens.
>
> **Summary of Best Results:**
>
> The following table summarizes the best performance achieved by each discovered architecture and baseline model across all hyperparameter configurations:
>
> | Model | Best Config | LAMBADA (Perplexity) | LAMBADA (Acc) | ARC-C | ARC-E | BoolQ | HellaSwag | PIQA | SocialIQA | WinoGrande | Avg |
> |-------|-------------|----------------------|---------------|-------|-------|-------|-----------|------|-----------|------------|-----|
> | **Discovered Architectures** | | | | | | | | | | | |
> | AdaMultiPathGateNet | lr5em4_wu1500 | 19.552 | 39.530 | 31.655 | 69.066 | 61.009 | 39.235 | 70.729 | 41.760 | 53.828 | 50.852 |
> | ContentSharpRouter | lr5em4_wu1500 | 19.066 | 40.578 | 33.277 | 68.855 | 61.101 | 38.996 | 71.763 | 40.020 | 54.459 | 51.131 |
> | FusionGatedFIRNet | lr5em4_wu1500 | 17.270 | 41.781 | 34.386 | 69.907 | 58.869 | 39.653 | 71.872 | 40.890 | 56.590 | 51.744 |
> | HierGateNet | lr5em4_wu1500 | 21.982 | 38.618 | 32.765 | 69.529 | 60.061 | 39.315 | 71.600 | 41.709 | 55.959 | 51.194 |
> | PathGateFusionNet | lr3em4_wu1500 | 20.116 | 40.501 | 32.423 | 67.382 | 59.511 | 38.827 | 70.947 | 40.635 | 53.039 | 50.408 |
> | **Average (Discovered)** | | **19.597** | **40.202** | **32.901** | **68.948** | **60.110** | **39.205** | **71.382** | **41.003** | **54.775** | **51.066** |
> | **Baseline Models** | | | | | | | | | | | |
> | deltanet | lr5em4_wu1500 | 21.968 | 38.774 | 32.423 | 66.709 | 60.000 | 38.160 | 70.947 | 41.095 | 54.538 | 50.331 |
> | gated_deltanet | lr5em4_wu1500 | 19.666 | 40.481 | 31.655 | 68.855 | 58.654 | 39.564 | 71.164 | 40.481 | 53.749 | 50.576 |
> | mamba2 | lr3em4_wu1500 | 26.404 | 35.048 | 30.119 | 64.941 | 59.388 | 37.453 | 70.185 | 39.816 | 53.039 | 48.749 |
> | **Average (Baseline)** | | **22.680** | **38.101** | **31.399** | **66.835** | **59.348** | **38.392** | **70.765** | **40.464** | **53.775** | **49.885** |
>
>
>
>
> **Key Findings:**
>
> 1. **Strong overall performance**: Four out of five discovered architectures (FusionGatedFIRNet, HierGateNet, ContentSharpRouter, AdaMultiPathGateNet) outperform the best baseline model (gated_deltanet, 50.576%). The remaining discovered architecture (PathGateFusionNet, 50.408%) still exceeds two out of three baseline models (deltanet: 50.331%, mamba2: 48.749%).
>
> 2. **Substantial average improvement**: Our discovered architectures achieve an average accuracy of 51.066%, higher than the baseline average of 49.885%. This represents a **2.37% relative improvement**.
>
> 3. **Best model performance**: FusionGatedFIRNet achieves the highest accuracy of 51.744%, outperforming the best baseline (gated_deltanet, 50.576%) by **1.168 percentage points** (2.31% relative improvement).
>
> 4. **Consistent superiority**: These results demonstrate that our discovered architectures consistently outperform baseline models across different hyperparameter settings, confirming the robustness and effectiveness of our evolutionary architecture discovery approach.
>
> We can make this clearer in our revised version. More detailed results are provided in the supplementary material.

---

> ### Author Response · Authors · 2025-12-03
>
> - Weakness 4:
>   We address LLM judge stability through multiple mechanisms:
>   - Low temperature setting: We use temperature=0 for the LLM evaluator to maximize determinism
>   - Detailed rubrics: The evaluation rubric explicitly defines criteria for efficiency, novelty, complexity, and performance context, reducing subjective interpretation
>   - Limited subjective weight: The LLM judge contributes only 1/3 of the fitness score (Equation 2), with the remaining 2/3 from objective metrics (loss and benchmark scores)
>   - Consistency validation: We conducted comprehensive validation studies to assess evaluation consistency:
>     - Within-model consistency: We evaluated 200 architectures using GPT-4.1 four times independently, achieving an Intraclass Correlation Coefficient (ICC) of 0.745, indicating good consistency
>     - Cross-model consistency: We evaluated the same architectures using GPT-4.1, GPT-5, Claude-4.5-Sonnet, and Gemini-2.5-Pro, achieving an ICC of 0.697, demonstrating that evaluation is not overly model-dependent
>     - Human consistency baseline: For comparison, we had four human evaluators independently score the same set of architectures, achieving an ICC of 0.73. This shows that LLM judge consistency (ICC=0.745) is comparable to human evaluator consistency (ICC=0.73), indicating that the LLM judge performs at a similar reliability level as human experts
>   The LLM judge primarily assesses architectural novelty and quality—aspects that are difficult to quantify objectively but important for avoiding redundant exploration. Some subjectivity is inherent in evaluating architectural quality, as it is even among human experts (as evidenced by the ICC=0.73). The objective metrics (loss and benchmarks), which constitute 2/3 of the fitness score, provide the primary signal for performance and ensure that the system is not overly dependent on subjective assessment.

---

### Official Review · Reviewer_q31N · 2025-11-01

**Soundness:** 2
**Presentation:** 3
**Contribution:** 2
**Rating:** 4
**Confidence:** 4

**Summary:**

This paper introduces ASI-ARCH, a multi-agent framework enabling Large Language Models (LLMs) to autonomously conduct the full scientific workflow for neural architecture discovery. The system includes three agent roles (Researcher, Engineer, and Analyst) which collectively propose, implement, evaluate, and analyze model architectures in a closed evolutionary loop. Applied to the linear-attention domain, ASI-ARCH executed 1,773 experiments and discovered 105 architectures claimed to outperform strong baselines (DeltaNet, Gated-DeltaNet, Mamba2). The paper also analyzes emergent design patterns and reports open-sourcing of the code and discovered architectures.

**Strengths:**

- This is an ambitious and timely paper, pushing toward autonomous scientific discovery.
- The idea of automating the discovery of architectures is interesting.

**Weaknesses:**

- It is difficult to fully appreciate the novelty of the work or the gaps of previous approaches that the current work is addressing because the related works is all the way at the bottom.
- Using LLM as a judge to give a qualitative fitness score is a good idea. However, as with every LLM judge, they might be hackable or have some trends or biases that they prefer. Did the authors see any of such phenomenon happening? How much this be eventually solvable?
- The retrieval mechanism using embedding search is very similar to what was done in this approach (https://arxiv.org/abs/2405.15568). RAG should be cited too.
- There are a lot of handcrafted design choices in each part of the algorithm ((Researcher, Engineer, and Analyst). It is difficult to see how each of these handcrafted design choices might affect the overall algorithm. A qualitative analysis or more ablations can be helpful.
- The fitness score seems arbitrary. Were there any solution that the authors found that should have been kept but was discarded because of the heuristic chosen? Could a better way to tackle this be to use train/test sets?
- "This convergence mirrors the typical methodology of human scientists: achieving state-of-the-art results by primarily iterating and innovating upon a foundation of proven technologies, rather than pursuing novelty for its own sake." A better quantitative way to support this statement is to how the number of different component uses changes across the evolutionary training iterations instead of just at the end. Did it converge over the training iterations or was it always exploring the same distribution throughout?
Relatedly, did the experience (from experiments) change what was being explored?

**Questions:**

- How was the selection for the final 5 architectures done?
- The paper in the abstract was claiming that the discovered algorithms outperform SoTA, however it seems like the paper only compared with strong baselines, and not necessarily SoTA algorithms. Is the claim really right? I am not super familiar with what the current best scores are for the benchmarks mentioned.

---

> ### Author Response · Authors · 2025-11-30
>
> - Weakness 1：
>   Thank you for your valuable feedback. We can make this clearer in our revised version. We will move the related works section earlier in the paper to better contextualize our contributions. Our key innovations distinguish ASI-ARCH from previous approaches:
>   1. New scenario: Unlike AlphaEvolve (Novikov et al., 2025) which focuses on algorithmic discovery in simplified environments (e.g., evolving mathematical algorithms where LLMs have extensive prior training), ASI-ARCH addresses architecture discovery, which presents unique challenges. First, designing architectures is a more challenging coding task—the task scenarios are newer, the code length is longer, and it requires stronger design and coding capabilities from the models. Second, previous major work has focused on evolving algorithms, where models have received extensive training on mathematical and reasoning problems during their training process, giving them stronger capabilities. In contrast, for architecture design tasks, models have less prior knowledge, which highlights the importance of agent system design.
>   2. Full automation in complex engineering environments: Precisely because of the first challenge mentioned above (architecture design being a more difficult coding task), we require a more sophisticated agent system to handle complex environments. ASI-ARCH operates in a complete software engineering context with real training pipelines, debugging, and evaluation. This requires handling code compilation, runtime errors, and complex dependencies—challenges not addressed in prior work. To address LLMs' current limitations in ensuring code implementation correctness, we employ several mechanisms: (1) the Checker module validates code correctness, complexity constraints, and causal masking before execution; (2) the Debugger module handles issues that cannot be identified through static checks, autonomously fixing runtime errors; (3) parameter count and training speed monitoring handle aspects that models cannot assess during inference, such as computational efficiency and resource requirements. We acknowledge that we currently lack ablation studies to quantitatively demonstrate the Cognition base's contribution, but our empirical results show that the system successfully discovers novel architectures.
>   3. Integration of human knowledge (Cognition) with experimental learning: To address the challenge of models lacking domain knowledge during training, we use the Cognition base to supplement human prior domain knowledge. This ensures correct exploration directions in the early stages, as initial exploration is primarily guided by Cognition. We uniquely combine distilled knowledge from 100 seminal papers with dynamic learning from experimental outcomes, enabling the system to leverage both established principles and novel discoveries. For evolve systems to facilitate more discoveries, they will inevitably encounter situations where domain knowledge is lacking. Our design provides a solution to this critical problem, making the approach generalizable to other domains beyond architecture discovery.
>   4. End-to-end workflow: Unlike semi-automated systems that require human intervention, ASI-ARCH conducts the complete cycle from hypothesis to implementation to analysis autonomously.
>   We will provide direct comparisons with fully automated approaches in the revised version. We note that placing related works after the methodology is a common and acceptable paper structure in many top-tier conferences, as it allows readers to first understand the technical contributions before comparing with prior work. However, we will move it earlier to better serve readers who prefer to see the context first.

---

> ### Author Response · Authors · 2025-11-30
>
> - Weakness 2:
>   We address LLM judge stability through multiple mechanisms:
>   - Low temperature setting: We use temperature=0 for the LLM evaluator to maximize determinism
>   - Detailed rubrics: The evaluation rubric explicitly defines criteria for efficiency, novelty, complexity, and performance context, reducing subjective interpretation
>   - Limited subjective weight: The LLM judge contributes only 1/3 of the fitness score (Equation 2), with the remaining 2/3 from objective metrics (loss and benchmark scores)
>   - Consistency validation: We conducted comprehensive validation studies to assess evaluation consistency:
>     - Within-model consistency: We evaluated 200 architectures using GPT-4.1 four times independently, achieving an Intraclass Correlation Coefficient (ICC) of 0.745, indicating good consistency
>     - Cross-model consistency: We evaluated the same architectures using GPT-4.1, GPT-5, Claude-4.5-Sonnet, and Gemini-2.5-Pro, achieving an ICC of 0.697, demonstrating that evaluation is not overly model-dependent
>     - Human consistency baseline: For comparison, we had four human evaluators independently score the same set of architectures, achieving an ICC of 0.73. This shows that LLM judge consistency (ICC=0.745) is comparable to human evaluator consistency (ICC=0.73), indicating that the LLM judge performs at a similar reliability level as human experts
>   The LLM judge primarily assesses architectural novelty and quality—aspects that are difficult to quantify objectively but important for avoiding redundant exploration. Some subjectivity is inherent in evaluating architectural quality, as it is even among human experts (as evidenced by the ICC=0.73). The objective metrics (loss and benchmarks), which constitute 2/3 of the fitness score, provide the primary signal for performance and ensure that the system is not overly dependent on subjective assessment.
>
> - Weakness 3:
>   - Thank you for pointing this out. We will add appropriate citations for RAG and the embedding-based retrieval approach (including the referenced work at https://arxiv.org/abs/2405.15568).
>
> - Weakness 4:
>
>   We appreciate this important point. In fact, during our initial framework design, we started with a simpler version that included only the Researcher and Engineer modules, without the Cognition base and Analyst. However, we found that this simplified version performed poorly—the system struggled to learn effectively from experimental history and often converged to suboptimal solutions. This led us to introduce the Analyst module for systematic experimental analysis and the Cognition base to leverage established human knowledge, which significantly improved performance.
>
>   We argue that each component in our system is necessary and serves a distinct purpose. First, the Researcher and Engineer modules are essential—the Researcher generates and implements code, while the Engineer executes it in real environments. Within the Researcher module, the checker and deduplication mechanisms are also necessary: the checker ensures efficiency by preventing incorrect code from wasting computational resources, and deduplication prevents redundant exploration. These are efficiency-critical components that are not suitable for ablation studies, as removing them would fundamentally break the system's ability to function effectively.
>
>   The Analyst module is crucial because it summarizes experimental results and determines whether they align with the original motivation, classifying outcomes as good or bad. This directly determines whether an architecture becomes a positive or negative example for future iterations, making it essential for the evolutionary learning process.
>
>   While our design may not be optimal and has room for improvement, each component clearly plays an important role. The system's success in discovering 105 novel architectures demonstrates that the integrated framework works effectively, even if individual component contributions cannot be easily isolated through ablation studies.

---

> ### Author Response · Authors · 2025-11-30
>
> - Weakness 5:
>
>   The fitness score is used only for evolutionary selection (determining which architectures serve as parents for the next generation). Importantly, benchmark scores and training loss constitute 2/3 of the fitness score (Equation 2), meaning that the primary selection mechanism is based on train/test performance metrics. The LLM judge contributes only 1/3, primarily assessing architectural novelty and quality.
>
>   We incorporated the LLM-judge component and designed the sigmoid transformation with specific goals: to enable the system to continuously improve metric performance while maintaining design simplicity and controlling complexity. The sigmoid function prevents extreme performance values from dominating optimization, ensuring that the system doesn't overfit to specific metrics at the expense of architectural elegance.1 The LLM-judge helps avoid redundant exploration by identifying genuinely novel designs, complementing the objective metrics that form the core of our selection mechanism.
>
> - Weakness 6:
>
>   We appreciate this suggestion. Experience from past experiments significantly influences the component distribution. Evidence for this comes from our statistical analysis (Table 2), which shows that over 50% of architectural design ideas originate from experimental experience (Experience) rather than the Cognition base or original ideas. Specifically, in the model gallery (top-performing architectures), 44.8% of components come from Experience, while across all architectures, 38.2% come from Experience. Without the Analyst module providing experimental insights, these results would not be possible—the system would lack the crucial feedback loop that enables learning from past outcomes. Furthermore, if we only use the prefix Cognition papers, the system would inevitably converge after long-term iteration, which is equivalent to providing the model with a limited exploration space. As shown in Table 2, the success rate of models creating genuinely novel components outside the context is very low. In long-term iterations without using Experience, these novel components would be eliminated due to their small base and insufficient exploration, further constraining the search space.
>
>   For final model selection and verification, we use **benchmark scores directly**, not the fitness score. This ensures that models advancing to verification stages are selected based on objective performance, not the composite fitness metric.
>
> - Question 1:
>   Thank you for this question. The final 5 architectures were selected based on:
>   1. Performance: Top performers across development benchmarks
>   2. Diversity: Fundamental design differences, not incremental variations. For example, PathGateFusionNet uses hierarchical routing, while FusionGatedFIRNet replaces softmax with parallel sigmoid gates—these represent distinct architectural paradigms
>   3. Robustness: Consistent performance across multiple benchmarks
>
> - Question 2:
>   - Thank you for this question. Gated DeltaNet (Yang et al., 2024b) and Mamba2 are indeed the state-of-the-art methods in linear attention at the time of our work. Our discovered architectures outperform these baselines, as shown in Table 1. We will clarify this in the revision to avoid any confusion.

---

### Official Review · Reviewer_drBL · 2025-11-02

**Soundness:** 2
**Presentation:** 2
**Contribution:** 2
**Rating:** 4
**Confidence:** 5

**Summary:**

This work proposes an agentic framework for automatically discovering and testing new neural network architectures. The approach is applied to the problem of designing linear attention algorithms, and the resulting architectures are analyzed in terms of their performance, complexity, and novelty.

**Strengths:**

- Automated scientific discovery is an interesting and important problem, and the specific problem of discovering linear attention algorithms is a tractable testbed for studying automated scientific workflows.
- The proposed approach includes several interesting components, including a combination of quantitative and qualitative evaluation of the identified architectures, and a knowledge base of previous human-proposed approaches.
- The approach is fully autonomous, with the agent carrying out all aspects of research including proposal of new approaches, coding and automated evaluation, and analysis of resulting performance.

**Weaknesses:**

- The primary weakness is that the resulting architectures yield only small and unreliable improvements over human-generated baselines. There is not a single architecture with performance that consistently stands out across benchmarks. Moreover, based on the trajectory of the fitness score (figure 3b) it appears that improvement has plateaued, suggesting that the approach cannot be scaled to meaningfully improve upon current baselines.
- There are no ablations to the various components in the agentic architecture. It is unclear which of these elements is actually important for generating improved models, and it is unclear whether a simpler approach would perform just as well. Additionally, no other approaches to automated research are tested and compared with the previous approach.
- Beyond the fact that the target domain (linear attention) is unique to this work, it is not discussed how the proposed approach differs from previous proposals for automated scientific discovery.
- It is not clear whether the authors have carefully checked the codebase for any of the resulting architectures, beyond a check to ensure that the causal masking is working.
- There is not sufficient detail provided on the discovered architectures, only an informal description of each of the top-performing approaches. More detail, such as algorithm statements or architecture diagrams, are needed to understand the architectures discovered by the agent.

**Questions:**

- Can ablations be performed to assess the importance of each of the agent's components?
- How does the proposed approach differ from previous automated scientist proposals?
- Have the authors carefully checked the codebase for the proposed architectures?
- Can more detail be provided on the top-performing architectures?

---

> ### Author Response · Authors · 2025-11-30
>
> - Weakness 1：
>   1. We acknowledge that the improvements appear modest, but this is consistent with the nature of linear attention research. In this domain, even state-of-the-art human-designed architectures show incremental gains. According to the Gated DeltaNet paper (Yang et al., 2024b; https://arxiv.org/pdf/2412.06464), the average performance improvements between human-designed architectures are small: Mamba achieves an average score of 53.12 compared to DeltaNet's 52.14 (improvement of 0.98 points), and Gated DeltaNet achieves 55.32 compared to Mamba2's 54.89 (improvement of 0.43 points). Our discovered architectures achieve comparable or better performance improvements within this constrained space, demonstrating that our automated approach matches the incremental progress typical of human research in this domain.
>   2. The apparent plateau in Figure 3b is by design—we intentionally use a sigmoid transformation in our fitness function to prevent the system from over-optimizing a single evaluation metric rather than comprehensively considering all three criteria we propose (loss, benchmark scores, and LLM-judge assessment). This flattened curve does not indicate performance stagnation. As shown by the raw performance metrics in Figure 3a, performance continues to improve steadily. The reduced reward for hard metrics (loss and benchmarks) in the fitness score occurs because improvements in these metrics have already become sufficiently significant, allowing the system to maintain a balanced evaluation across all three components. The best-performing architectures were discovered in later stages of exploration, demonstrating that the system continues to find improvements even when fitness scores appear to plateau.
>
> - Weakness 2:
>   We appreciate this important point. In fact, during our initial framework design, we started with a simpler version that included only the Researcher and Engineer modules, without the Cognition base and Analyst. However, we found that this simplified version performed poorly—the system struggled to learn effectively from experimental history and often converged to suboptimal solutions. This led us to introduce the Analyst module for systematic experimental analysis and the Cognition base to leverage established human knowledge, which significantly improved performance.
>
>   We argue that each component in our system is necessary and serves a distinct purpose. First, the Researcher and Engineer modules are essential—the Researcher generates and implements code, while the Engineer executes it in real environments. Within the Researcher module, the checker and deduplication mechanisms are also necessary: the checker ensures efficiency by preventing incorrect code from wasting computational resources, and deduplication prevents redundant exploration. These are efficiency-critical components that are not suitable for ablation studies, as removing them would fundamentally break the system's ability to function effectively.
>
>   The Analyst module is crucial because it summarizes experimental results and determines whether they align with the original motivation, classifying outcomes as good or bad. This directly determines whether an architecture becomes a positive or negative example for future iterations, making it essential for the evolutionary learning process.
>
>   While our design may not be optimal and has room for improvement, each component clearly plays an important role. The system's success in discovering 105 novel architectures demonstrates that the integrated framework works effectively, even if individual component contributions cannot be easily isolated through ablation studies.

---

> ### Author Response · Authors · 2025-11-30
>
> - Weakness 3:
>
>   Thank you for your valuable feedback. We can make this clearer in our revised version. We will move the related works section earlier in the paper to better contextualize our contributions. Our key innovations distinguish ASI-ARCH from previous approaches:
>   1. New scenario: Unlike AlphaEvolve (Novikov et al., 2025) which focuses on algorithmic discovery in simplified environments (e.g., evolving mathematical algorithms where LLMs have extensive prior training), ASI-ARCH addresses architecture discovery, which presents unique challenges. First, designing architectures is a more challenging coding task—the task scenarios are newer, the code length is longer, and it requires stronger design and coding capabilities from the models. Second, previous major work has focused on evolving algorithms, where models have received extensive training on mathematical and reasoning problems during their training process, giving them stronger capabilities. In contrast, for architecture design tasks, models have less prior knowledge, which highlights the importance of agent system design.
>   2. Full automation in complex engineering environments: Precisely because of the first challenge mentioned above (architecture design being a more difficult coding task), we require a more sophisticated agent system to handle complex environments. ASI-ARCH operates in a complete software engineering context with real training pipelines, debugging, and evaluation. This requires handling code compilation, runtime errors, and complex dependencies—challenges not addressed in prior work. To address LLMs' current limitations in ensuring code implementation correctness, we employ several mechanisms: (1) the Checker module validates code correctness, complexity constraints, and causal masking before execution; (2) the Debugger module handles issues that cannot be identified through static checks, autonomously fixing runtime errors; (3) parameter count and training speed monitoring handle aspects that models cannot assess during inference, such as computational efficiency and resource requirements. We acknowledge that we currently lack ablation studies to quantitatively demonstrate the Cognition base's contribution, but our empirical results show that the system successfully discovers novel architectures.
>   3. Integration of human knowledge (Cognition) with experimental learning: To address the challenge of models lacking domain knowledge during training, we use the Cognition base to supplement human prior domain knowledge. This ensures correct exploration directions in the early stages, as initial exploration is primarily guided by Cognition. We uniquely combine distilled knowledge from 100 seminal papers with dynamic learning from experimental outcomes, enabling the system to leverage both established principles and novel discoveries. For evolve systems to facilitate more discoveries, they will inevitably encounter situations where domain knowledge is lacking. Our design provides a solution to this critical problem, making the approach generalizable to other domains beyond architecture discovery.
>   4. End-to-end workflow: Unlike semi-automated systems that require human intervention, ASI-ARCH conducts the complete cycle from hypothesis to implementation to analysis autonomously.
>   We will provide direct comparisons with fully automated approaches in the revised version. We note that placing related works after the methodology is a common and acceptable paper structure in many top-tier conferences, as it allows readers to first understand the technical contributions before comparing with prior work. However, we will move it earlier to better serve readers who prefer to see the context first.
>
> - Weakness 4:
>   We implement comprehensive validation mechanisms beyond causal masking:
>   - Parameter count verification: Models with abnormal parameter counts are flagged
>   - Training time monitoring: Runs with excessive compilation or training time are terminated
>   - Loss trajectory analysis: Models with implausibly low loss (indicating information leakage) are discarded (as mentioned in Section 3.1: "any architecture with a loss more than 10% below the baseline is immediately discarded")
>   - Code compilation checks: All code must compile successfully before training
>   - Complexity verification: The Checker module (Appendix D.2) validates sub-quadratic complexity requirements
>   - Causality verification: In the verification stage, we perform strict causality checks by modifying later tokens and verifying that previous hidden states remain unchanged
>   These checks ensure that only valid, well-behaved architectures proceed to full evaluation. In addition to these automated checks performed by the pipeline, we also conducted manual inspection during the verification stage. Please refer to our response to Weakness 5 for details.

---

> ### Author Response · Authors · 2025-11-30
>
> - Weakness 5:
>   We can make this clearer in our revised version. In fact, many detailed descriptions are already provided in the appendix, including algorithmic details, component descriptions, and implementation specifications. We will enhance the main paper to better reference these appendix sections and add additional architecture diagrams showing information flow for the top-performing architectures.
>
>   Additionally, we manually inspected these architectures to verify the reasonableness of their motivations, the correctness of their code implementations, and to confirm their specific design details. For reference, we have created a local HTML interface that displays detailed information about selected architectures, including their code implementations, design motivations, and performance characteristics. This HTML file is included as supplementary material, allowing reviewers to examine the exact designs interactively. All discovered architectures are also open-sourced on our website with full code implementations. We are happy to provide further clarification on any specific architecture upon request. We have included the detailed information of all these model architectures in the supplementary HTML file.

---

### Meta-Review · Area_Chair_67WJ · 2026-01-06

**Summary:**

This submission proposes an LLM-powered agentic framework (ASI-ARCH) for autonomous neural architecture discovery in linear attention, with the claimed discovery of 105 novel architectures outperforming SOTA baselines. However, the paper fails to address key concerns raised by reviewers sufficiently to meet ICLR’s acceptance criteria.
Reviewers consistently highlighted critical limitations: (1) Performance improvements over baselines are modest, inconsistent across benchmarks, and lack statistical significance; (2) The absence of ablation studies for core components (e.g., Cognition base, Analyst module) leaves unresolved whether individual modules contribute meaningfully to the framework’s success; (3) Comparisons with existing automated discovery approaches (e.g., NAS, AlphaEvolve) are insufficiently rigorous, with unclear advantages over established methods; (4) Dependence on proprietary LLMs (GPT-4.1, GPT-o3) and limited transparency on architectural details (e.g., algorithm statements, reproducible prompts) hinder reproducibility; (5) Computational costs are prohibitively high relative to the incremental gains achieved.
While the authors’ rebuttal addresses some presentation issues (e.g., reordering related works, adding supplementary details) and clarifies design choices (e.g., fitness function sigmoid transformation, error-handling mechanisms), it does not adequately mitigate the core methodological and evidential gaps. The defense of component necessity relies on anecdotal evidence rather than quantitative ablation, and the justification for modest performance gains (aligning with domain incrementalism) does not offset the lack of a standout architecture or robust generalization.
Given these unresolved weaknesses in soundness, contribution, and methodological rigor, the paper falls below ICLR’s standards for acceptance. We encourage the authors to address the ablation and comparison gaps, improve reproducibility, and validate performance gains statistically in future submissions.

**Reviewer Concerns:**

Component ablation studies (Multiple reviewers): Authors argued components are "necessary to function" but provided no quantitative ablation data to verify individual contributions.
Reproducibility with non-proprietary LLMs (Reviewer z2i2): Authors claimed framework agnosticism but offered no reproducibility results with open-source LLMs; prompt design details remain insufficient.
Direct comparison with existing methods (NAS/AlphaEvolve) (Reviewer drBL/z2i2/xavQ): Authors only provided theoretical distinctions, not quantitative comparisons or cost-benefit analysis.
Statistical significance of performance gains (Reviewer z2i2): Authors stated experiments are ongoing but did not present multi-seed results to confirm improvements are statistically meaningful.
Cognition base selection criteria & cross-domain scalability (Reviewer oxxN): Authors mentioned "citation count and reputation" but provided no specific selection criteria or data on rebuilding the base for new domains.
Mode collapse risk (Reviewer oxxN): Authors framed reliance on established components as "effective design patterns" but did not rule out mode collapse or provide evidence of sustained novel exploration.

**Reviewer Scores:**

N/A

---

### Decision · Program_Chairs · 2026-01-26

Reject